# The p3 peptides (Aβ$_{17-40/42}$) rapidly form amyloid fibrils that cross-seed with full-length Aβ

Yao Tian[1], Andrea P. Torres-Flores[1], Qi Shang[1], Hui Zhang[1], Anum Khursheed[1], Bogachan Tahirbegi[1], Patrick N. Pallier [2] & John H. Viles [1]✉

The p3 peptides, Aβ$_{17-40/42}$, are a common alternative cleavage product of the amyloid precursor protein, and are found in diffuse amyloid deposits of Alzheimer's and Down Syndrome brains. The p3 peptides have been mis-named 'non-amyloidogenic'. Here we show p3$_{40/42}$ peptides rapidly form amyloid fibrils, with kinetics dominated by secondary nucleation. Importantly, cross-seeding experiments, with full-length Aβ induces a strong nucleation between p3 and Aβ peptides. The cross-seeding interaction is highly specific, and occurs only when the C-terminal residues are matched. We have imaged membrane interactions with p3, and monitored Ca$^{2+}$ influx and cell viability with p3 peptide. Together this data suggests the N-terminal residues influence, but are not essential for, membrane disruption. Single particle analysis of TEM images indicates p3 peptides can form ring-like annular oligomers. Patch-clamp electrophysiology, shows p3$_{42}$ oligomers are capable of forming large ion-channels across cellular membranes. A role for p3 peptides in disease pathology should be considered as p3 peptides are cytotoxic and cross-seed Aβ fibril formation in vitro.

Alzheimer's disease (AD) accounts for at least two thirds of dementias, with currently 50 million sufferers worldwide[1]. AD is characterised by the accumulation of peptides cleaved from the amyloid precursor protein (APP), which form diffuse amyloid deposits and senile plaques in the brain[2]. Inherited forms of AD are caused by mutations in APP, or the γ-secretase responsible for APP cleavage, this makes a strong causal link between peptides cleaved from APP and Alzheimer's disease[3–6].

Aβ and p3 are endogenous peptides that are cleaved from the larger transmembrane amyloid precursor protein (APP) by the action of three enzymes, Fig. 1[5–7]. The β-secretase cleaves Aβ at its N-terminus. While the γ-secretase complex, cleaves APP, to form the C-terminus of Aβ. The cleavage point is variable but typically results in Aβ peptide 40 or 42 amino acids in length. An alternative cleavage pathway caused by the action of the α-secretase can result in a much shorter more hydrophobic peptide, named p3, which contains residues 17-40/42 of

the Aβ sequence[8,9]. To distinguish it from the β-cleavage, this pathway has been mis-named 'non-amyloidogenic'. Using cultured cortical neurons, it has been shown APP is cleaved by α-secretase twice as frequently as by the β-secretase[10].

The p3 peptide is reported to be found in selected areas of the AD brain, in particular, diffuse amyloid deposits and at low levels in senile plaques[11,12]. The p3 peptides have also been identified in cerebrospinal fluid for AD patients[13]. While for those with Down syndrome, which is associated with a form of early-onset AD, the p3 peptide is dominant over full-length Aβ in pre-amyloid deposits of the cerebellum[14].

Perhaps because of its designation as non-amyloidogenic and its poor solubility, there have been few biophysical studies of p3, relative to full-length Aβ. The contrast in publication rate is highlighted in Supplementary Fig. 1. For a careful review of these studies see; Kuhn and Raskatov[15]. Early studies on p3 report conflicting observations, with some studies indicating p3 does not form oligomers[16] or fibrils[17],

[1]Department of Biochemistry, School of Biological and Behavioural Sciences, Queen Mary University of London, London E1 4NS, UK. [2]The Blizard Institute, Centre for Neuroscience, Surgery and Trauma, Queen Mary University of London, London E1 2AT, UK. ✉e-mail: j.viles@qmul.ac.uk

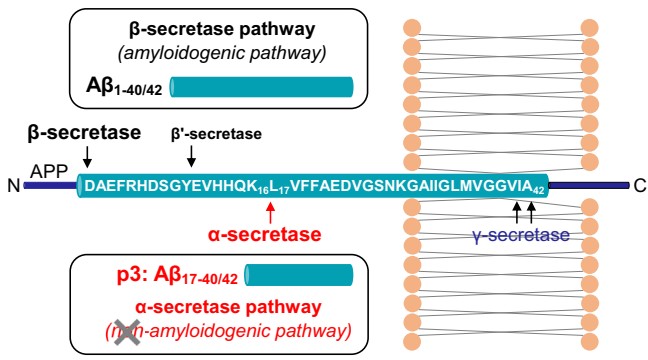

**Fig. 1 | β- and α-secretase processing of amyloid precursor protein (APP).** The amyloidogenic pathway, when Aβ is produced by the sequential cleavages of APP by β- and γ- secretases. The alternative pathway, mis-named as non-amyloidogenic, in which APP is cleaved by α-secretase between Lys16 and Leu17, to produce p3. γ-Secretase cleaves the C-terminus most commonly at position 40 or 42, within the lipid bilayer.

while others have reported the opposite[18-20]. These discrepancies may be down to the difficulty in handling this poorly soluble peptide. More recent studies have confirmed $p3_{40}$ can indeed form amyloid fibrils, while $p3_{42}$ was found to be difficult to solubilize. Even in recent studies, the lag-phase in the kinetics of $p3_{40}$ fibril formation was not observed, while the $p3_{42}$ fibril growth kinetics were not studied[21,22].

As both p3 and Aβ peptides are present in diffuse amyloid deposits and senile plaques, there is a possibility they impact each other's fibril formation kinetics. Self-seeding is a well-established phenomenon, where preformed fibrils of the same Aβ peptide sequence nucleate fibril formation, this circumvents primary nucleation processes and reduces fibril formation lag-times[23,24]. While the 'parent' seeds often propagates the same structural morphology in the 'daughter' fibril[25]. Potentially cross-seeding, where different sequences of Aβ might nucleate fibril formation may also occur in vivo. Thus, a relatively small proportion of a highly amyloidogenic Aβ isoform, could have a major impact on fibril formation kinetics for other more abundant isoforms. There have been relatively few studies of this cross-seeding phenomena, despite its physiological relevance, and these have been largely restricted to studies of cross-seeding between $Aβ_{40}$ and $Aβ_{42}$[23,26-28]. It is becoming clear this process is very peptide specific[29-31] and should not be confused with a less specific oligomer interaction before fibril formation[23], or accumulation of plaques once fibrils are formed[32]. It is generally understood $Aβ_{40}$ and $Aβ_{42}$ do not cross-fibrillize but form fibrils independently[23,26-28]. The loss of just two amino acids from the C-terminus makes $Aβ_{40}$ and $Aβ_{42}$ fibril structures incompatible. Preliminary studies have suggested cross-seeding between $p3_{40}$ and $Aβ_{40/42}$[21,22], while cross-seeding between $p3_{42}$ and $p3_{40}$ and the cross-seeding between $p3_{42}$ and $Aβ_{40/42}$ have yet to be explored. More widely, there are numerus examples of the interaction and cross-seeding between different amyloid forming proteins, which may impact fibril formation in-vivo[33-35].

The mechanism by which Aβ and perhaps p3 triggers the cascade of events that leads to AD is not yet fully resolved, one popular hypothesis surrounds the action of Aβ on lipid membranes[36-38]. Aβ oligomers and curvilinear protofibrils carpet the lipid membrane inserting into the upper leaflet[36], this disrupts the integrity of the membrane causing permeability and unregulated cellular $Ca^{2+}$ influx[39,40]. A more specific Aβ interaction with the bilayer involves ring-shaped annular assemblies, these can insert into the membrane forming large unregulated ion-channel pores[41-45]. This allows flow of $Ca^{2+}$ and other ions into the cell. This membrane permeability can then trigger a cascade of cellular events, including phosphorylation of tau, that culminates in a loss of neurons and dementia[3,4].

Here we aim to characterise the assembly properties of the p3 peptides; $Aβ_{(17-40/42)}$. We will investigate the cross-seeding kinetics of

p3 with full-length Aβ peptides in vitro. Furthermore, we will probe the ability of the p3 peptides to disrupt lipid membranes and cause $Ca^{2+}$ influx and cytotoxicity. We show p3 peptides form amyloid fibrils more rapidly than Aβ, and p3 can cross-seed and nucleate the fibril formation of full-length Aβ, but only for specific Aβ:p3 combinations with the same C-terminal length. The cytotoxic p3 peptides form annular oligomers, and $p3_{42}$ oligomers form ion-channel pores across cellular plasma membranes. The misnaming of p3 as non-amyloidogenic has meant its possible role in Alzheimer's disease has too often been underplayed. Indeed it is often suggested that elevating α-secretase activity might be protective by reducing levels of Aβ and so might be a therapeutic approach[9], however here we show production of p3 rather than Aβ may in fact be undesirable.

## Results

### Fibril formation, p3 kinetics is dominated by secondary nucleation

The amyloid specific dye, thioflavin-T (ThT), has been used to monitor p3 fibril formation. In contrast to previous studies that were unable to obtain fibril growth kinetic curves, with a clear lag-phase for $p3_{40}$ or $p3_{42}$[21,22], we have found by solubilizing $p3_{40/42}$ peptide in aqueous solution, at pH 10, followed by size-exclusion chromatography (SEC) purification, to remove any remaining nucleating seeds, it was possible to obtain highly reproducible kinetic data for both $p3_{40}$ and $p3_{42}$ fibril formation. We show the kinetic traces exhibit sigmoidal growth-curves characteristic of a nucleation polymerization reaction and exhibit a clear lag- and elongation- phase, Fig. 2 and Supplementary Fig. 2. The loss of the many solubilizing hydrophilic amino acids, within residues 1–16, results in the p3 peptides forming fibrils much more readily than full-length Aβ, under the same conditions. Indeed, the reaction half-time, $t_{50}$, are reduced by more than half, for the p3 peptides, Supplementary Fig. 2f. In contrast to previous reports[17,22], we also show that p3 and Aβ fibrils produce comparable ThT fibril fluorescence intensities, for preparations with the same initial monomer concentration, Supplementary Fig. 2e.

There are numerous steps that describe the molecular processes of assembly into amyloid fibrils. Each step has its own micro-rate constants, associated with the macroscopic kinetic profiles[46,47]. It has previously been shown that full-length Aβ is dominated by secondary nucleation[46], these processes are believed to involve the formation of nucleating oligomers on the surface of fibrils. We wondered if the $p3_{40/42}$ peptides were also dominated by secondary nucleation.

We have monitored p3 fibril formation for both $p3_{42}$ and $p3_{40}$, over a range of initial monomer concentrations, between 4-12.5 μM, Fig. 2 and Supplementary Figs. 3, 4. Figure 2c shows the reaction half-time ($t_{50}$) scales with the initial monomer concentration, under quiescent conditions. The scaling exponent (slope) are as follows: γ = -1.33 ($p3_{42}$) and γ = -1.39 ($p3_{40}$). These values are very similar to that reported for full-length $Aβ_{42}$ (also γ = -1.33)[46]. The linear dependence of this double logarithmic plot, Fig. 2c, indicates saturation effects are not important over the concentration range we have used[46-48]. We have used the online fitting programme, 'AmyloFit'[48] to globally fit the macro-kinetic traces for both $p3_{42}$ and $p3_{40}$, over a range of concentrations for the whole reaction course, Fig. 2d-f and Supplementary Fig. 3. Here we show that p3 fibril formation kinetic processes are indeed also dominated by secondary nucleation($k_2$). This is where nucleating oligomeric seeds form on the surface of fibrils[47,49,50]. The kinetics is both monomer and fibril concentration dependent, resulting in sharp exponential growth in the kinetic traces, which is not so marked when only primary nucleation takes place, Fig. 2d.

### p3 assemblies, secondary structure and morphology

Transmission electron microscopy (TEM) has been used to image p3 peptides at different time points during fibril assembly. Sampling at the end of the lag-phase of fibril growth, the images reveal prefibrillar

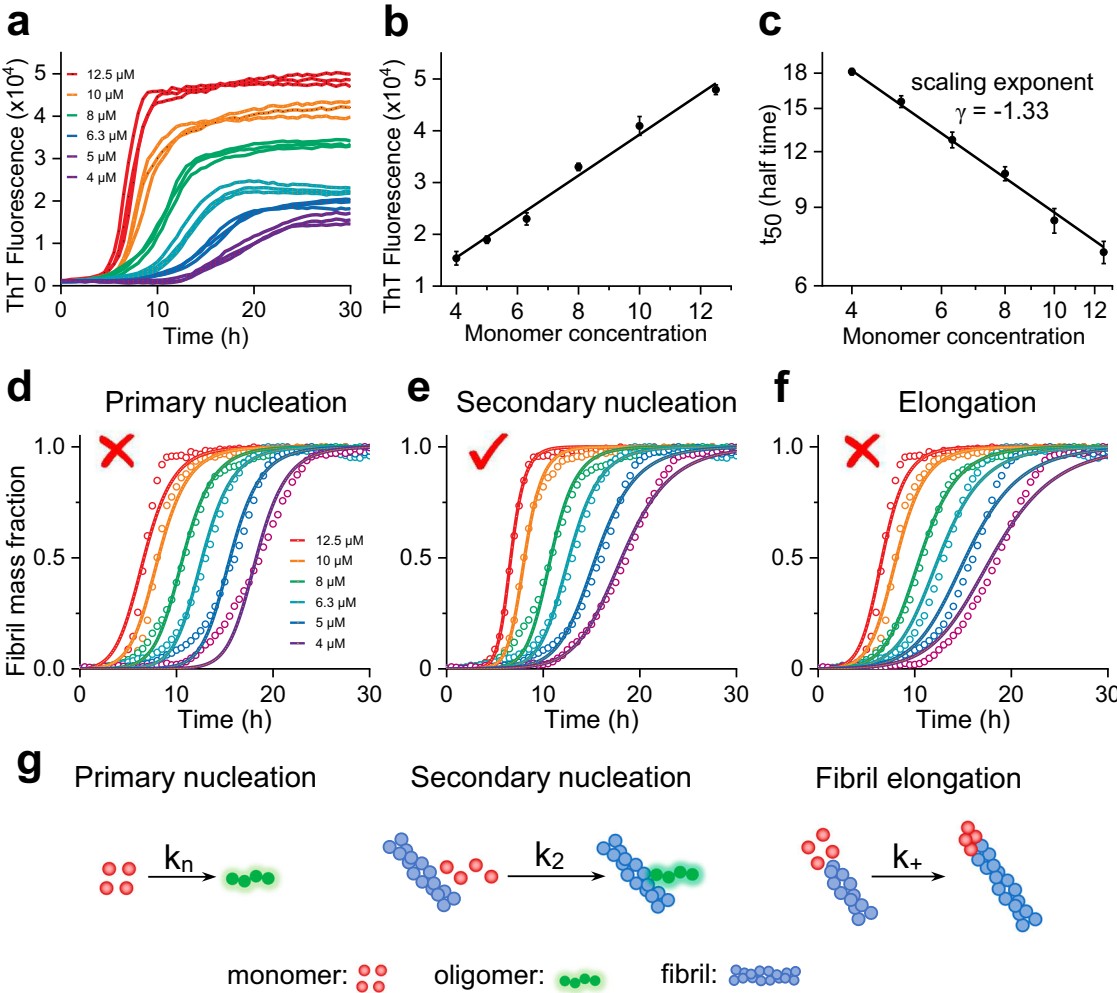

**Fig. 2 | Concentration dependent aggregation of p3₄₂. a** Kinetics profiles of $p3_{42}$ at initial monomer concentration from 4 µM (purple) to 12.5 µM (red) with individual technical replicates. **b** ThT fluorescence at the plateau-phase against the $p3_{42}$ monomer concentrations. **c** Double logarithmic plot of half time ($t_{50}$) versus the initial monomer concentrations, scaling exponent γ = -1.33. Error bars are standard error of the mean (SEM) from three replicates. **d**–**f** Global fits of the kinetic traces when only primary nucleation (**d**), secondary nucleation (**e**) and fibril elongation (**f**) rate constants are altered to globally fit concentration dependent traces. Kinetic traces fit well to secondary nucleation with a low mean residual error (MRE = 0.0009), while primary nucleation and fibril elongation fit less well (MRE = 0.0019; 0.0023 respectively). Global fitting indicates fibril formation is dominated by secondary nucleation. The 'check' and 'cross' marks on the charts refer to the kinetic traces fitting well or less well to the models. **g** Schemes of the microscopic steps for primary nucleation ($k_n$), secondary nucleation ($k_2$) and fibril elongation ($k_+$).

oligomeric and curvilinear protofibril structures, Fig. 3a, b. Additional, TEM images of oligomeric $p3_{40}$ and $p3_{42}$ are shown in Supplementary Fig. 5. After further incubation TEM images indicate fibrillar assemblies become widespread. The fibril morphology has a similar appearance to full-length Aβ. The twists in these fibrils, formed from SEC purified p3 peptide, under quiescent conditions, are compared to full-length Aβ fibrils, formed under the same conditions, Fig. 3c. Despite the loss of a third of the polypeptide chain, the fibril twist morphology of the p3 peptides are similar to the full-length Aβ counterparts, Fig. 3 and Supplementary Fig. 6. In particular, $A\beta_{40}$ and $p3_{40}$ have a similar pronounced twist periodicity, both have a long crossover distance within a range of between 101 and 166 nm for $A\beta_{40}$ and 106–134 nm for $p3_{40}$, Fig. 3c. While $A\beta_{42}$ and $p3_{42}$ counterparts have much tighter twists, with a crossover distance of just 30–38 nm and 41–71 nm, respectively. Furthermore, the range of fibril widths have a similar relationship. $A\beta_{40}$ and the $p3_{40}$ counterpart have similar fibril widths of 12–17 nm and 11–14 nm, while, $A\beta_{42}$ and $p3_{42}$ peptides have narrower widths of 7–10 nm and 8–12 nm, Fig. 3d. The mean fibril crossover distance and width are presented in Supplementary Table 1 for the four peptides. Histograms of the distribution of crossover distance and fibril widths

are shown in Supplementary Fig. 7. The range of fibril twists indicates some polymorphism for each peptide[51,52]. However, the range of values clearly group together, with a relatively long helical crossover lengths for $A\beta_{40}$ and $p3_{40}$ (134 nm and 120 nm) and much shorter crossover lengths for $A\beta_{42}$ and $p3_{42}$ (34 nm and 56 nm), Supplementary Table 1. It remains to be established how similar the fold topology is between these p3 and Aβ fibrils.

Circular dichroism (CD) spectra are consistent with a cross-β structure for the fibrils of p3 peptides, Fig. 3e, f. The CD spectra of the p3 fibrils are dominated by a strong negative CD band, centred at 217 nm and a positive band at 198 nm. This indicates a high proportion of β-sheet present. The SEC purified monomer is dominated by a negative band at 198 nm, which is characteristic of irregular structure, while lag-phase oligomers show an increased proportion of β-sheet, but still retain many monomers with a random-coil ellipticity in the mixed solution.

### Cross-seeding between p3 and Aβ peptides
The p3 and Aβ peptides are both released at the synapse and so may potentially influence the kinetics and morphology of each other's fibril

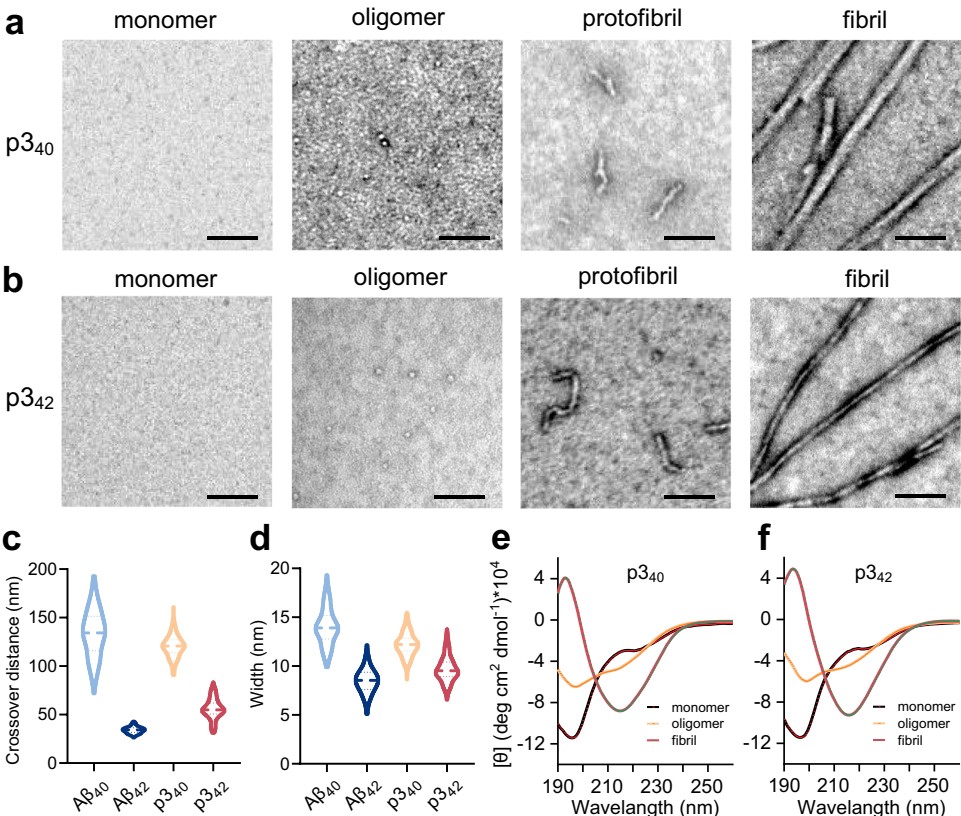

**Fig. 3 | The self-assembly of p3$_{40}$ and p3$_{42}$. a, b** TEM images of monomer, oligomer, protofibril and fibril of p3$_{40}$ (**a**) and p3$_{42}$ (**b**). The morphology of the p3 peptides were reproducible and consistent on at least three independent experiments. Scale bars: 50 nm. **c, d** Comparison of fibril twist periodicity (**c**) and fibril widths (**d**). Data suggest similar morphology for fibril with the same C-terminal truncation. **e, f** Far-UV CD spectra of p3$_{40}$ (**e**) and p3$_{42}$ (**f**) indicates transition in secondary structure from random coil (198 nm) to β-sheet (217 nm) for fibrils. Preparations (10 µM) were incubated in 10 mM sodium phosphate buffer, pH 7.4.

formation in vivo. The p3 peptides lack many of the solubilizing side-chains in the N-terminus of Aβ and consequently form fibrils much more rapidly, in half the time of full-length Aβ, under the same conditions, Supplementary Fig. 2. We wondered once p3 peptides form oligomers and fibrils, these assemblies might be capable of nucleating full-length Aβ fibril formation. In Fig. 4 and Supplementary Fig. 8, we show self- and cross- seeding experiments between p3 and Aβ. Very pronounced acceleration in fibril formation does indeed occur, with particular combinations of Aβ and p3 seeds. The nucleating interactions are very specific, with p3$_{40}$ fibril seeds (1 µM, 10% p3$_{40}$) nucleating Aβ$_{40}$ monomer and reducing lag-times by almost half, while p3$_{40}$ fibril seeds have almost no effect on the kinetics of p3$_{42}$ or Aβ$_{42}$, Fig. 4a. Similarly, only p3$_{42}$ is capable of cross-seeding with Aβ$_{42}$ monomers, Fig. 4b. This behaviour is precisely echoed with monomeric p3$_{40}$ and p3$_{42}$ while using fibril seeds of Aβ$_{40/42}$, Fig. 4c, d. The self- and cross-seeding pattern is very apparent when tabulated in Fig. 5. This bi-directional nature of cross-seeding is highlighted by the mirror symmetry across the diagonal shown in Fig. 5. Self-seeding typically causes a halving of lag-times (using 1 µM, fibril seeds), but also a very similar reduction in $t_{lag}$ and $t_{50}$ times for cross-seeding, but only between the p3$_{40}$:Aβ$_{40}$ and p3$_{42}$:Aβ$_{42}$ pairs, Fig. 5, Supplementary Table 2. Other combinations of p3 with full-length Aβ have negligible impact on kinetics and do not cross-seed, Fig. 5. Also of note, there is minimal cross-seeding between p3$_{40}$ and p3$_{42}$, Fig. 5, this is analogous to the minimal cross-seeding reported between Aβ$_{40}$ and Aβ$_{42}$[22].

We suggest the strong influence of p3$_{40/42}$ on the complementary fibril forms Aβ$_{40}$ or Aβ$_{42}$, relates to the impact the two C-terminal residues have on fibril structure. In particular, it is well documented the addition of two amino-acids have a major influence on the fundamental fold of Aβ$_{40}$ and Aβ$_{42}$. The two isoforms typically form an 'U-shaped/extended' and a 'S-shaped' folds respectively, these are apparent in many of the fibril structures reported under a variety of conditions[53–58]. The extension by two amino acids at the C-terminus, allows the stabilisation of the S-shaped fold, in which a salt-bridge is formed between the Ala42, C-terminal carboxylate, and the amino group of Lys28. This is not possible in Aβ$_{40}$ and so a salt-bridge is formed between Lys28 and Asp23[55,56]. The cross-seeding between p3 and Aβ suggests the C-terminus is fundamental to influencing the fold within fibrils and thus the compatibility between nucleating seeds. While the loss of residues 1–16 have less of an impact on the fold. We suggest that the structure of p3$_{40}$ and p3$_{42}$ will show similar U- and S-shaped topologies respectively. Although atomic details are yet to be resolved for the p3 peptides, the similarities in structure to the full-length Aβ$_{40}$ and Aβ$_{42}$ counterparts is suggested by the similarity in fibril twist periodicity and widths, shown in Fig. 3c, d and Supplementary Table 1.

It is notable the reduction in lag-times observed for cross-seeding is comparable in magnitude to the Aβ and p3 self-seeding kinetics, perhaps suggesting that the molecular processes are similar. It remains to be established if the cross-seeding phenomena is driven by fibril surface-catalysed secondary-nucleation or templating from the ends of seeding fibrils. In the case of self-seeding both lateral surface-catalysed secondary-nucleation and templating elongation occurs. Koloteva-Levine et al reports an excellent analysis of these two molecular behaviours[59].

We have previously studied a different N-terminal truncated form of Aβ; Aβ$_{11-40/42}$, formed from an alternative cleavage site of the β-secretase[29]. Like the pattern of cross-seeding for the p3 peptides in this study, we also showed that cross-seeding can occur only between Aβ$_{40}$ and Aβ$_{11-40}$, while Aβ$_{11-42}$ is only compatible with Aβ$_{42}$ and will cross-seed and cross-fibrilize[29]. A pattern of behaviour is emerging for this

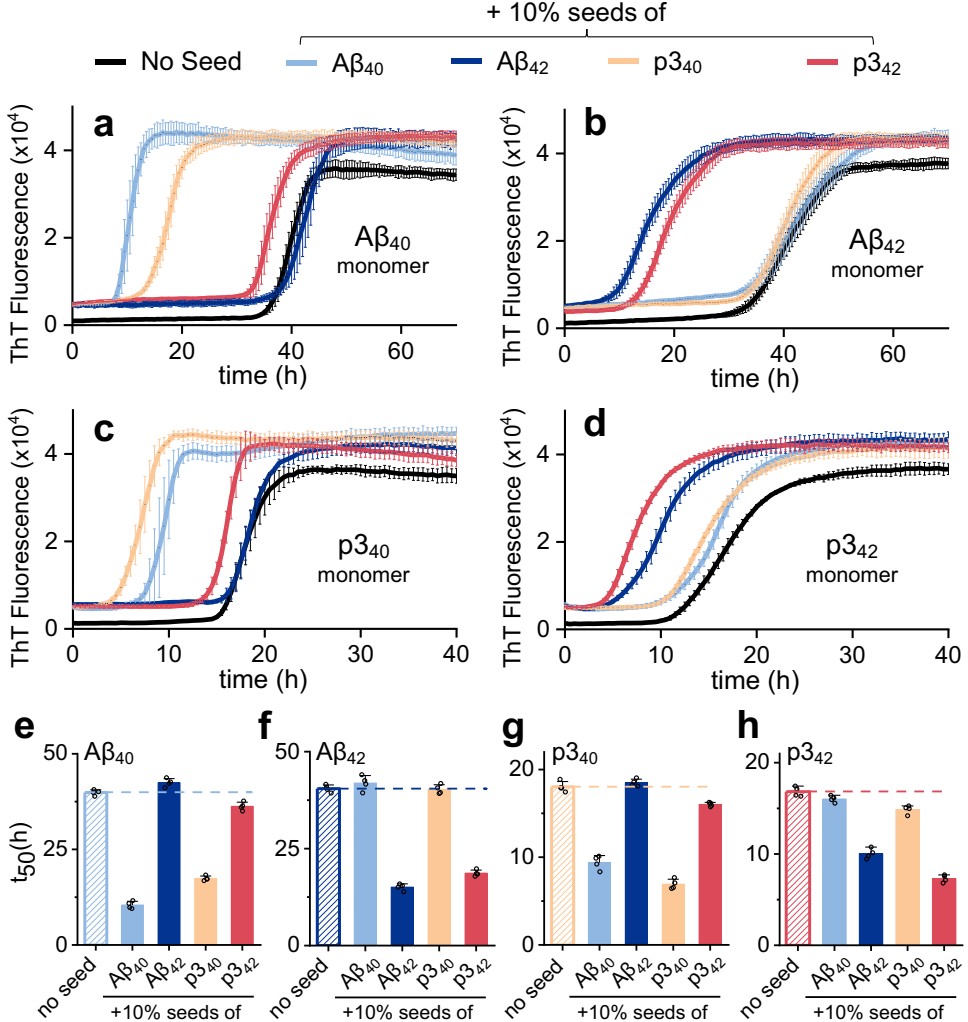

**Fig. 4 | Aβ cross-seeding with p3 peptides. a–d** Fibril formation of monomeric Aβ$_{40}$ (**a**), Aβ$_{42}$ (**b**), p3$_{40}$ (**c**) and p3$_{42}$ (**d**) in presence of a range Aβ isoform fibril seeds (10% w/w): No seed (black); Aβ$_{40}$ (pale blue); Aβ$_{42}$ (dark blue); p3$_{40}$ (orange); and p3$_{42}$ (red) seeds. **e–h** The mean $t_{50}$ are shown for Aβ$_{40}$ (**e**), Aβ$_{42}$ (**f**), p3$_{40}$ (**g**) and p3$_{42}$ (**h**) fibril formation in presence of a range Aβ isoform fibril seeds. Error bars are SEM from four replicates. Kinetic curves represent mean from individual technical replicates.

very specific cross-seeding behaviour. Deletions of residues 1–10 or 1–16 have little impact on the compatibility between Aβ peptides. While the C-terminal truncation of just two residues impacts the ability for Aβ to cross-seed. We have recently shown compatibility between fibril cross-seeding of Aβ$_{42}$ with the Arctic mutant Aβ$_{40}$(E22G), this is also associated with the U- and S-shaped fold, linked with the salt-bridge at Asp23-Lys28[30].

## p3 membrane interaction and cellular homeostasis

It is clear p3 peptides are capable of forming prefibrillar oligomeric and curvilinear protofibril structures, Fig. 3a,b. We therefore wondered if they might interact with lipid bilayers and cause membrane permeability, similar to that reported for full-length Aβ[36,37]. We have used large unilamellar vesicles (LUVs) and imaged the interaction of p3 peptides with lipid bilayers, and then compared the affects with their full-length Aβ counterparts. In Fig. 6, we show a series of typical LUVs, imaged by TEM, incubated in the presence of Aβ or p3 peptides. Lipid vesicles in buffer, negatively stained with uranyl-acetate, have a largely circular (spherical) appearance, with a relatively smooth membrane surface, Fig. 6a and Supplementary Fig. 9, 10. Oligomers and curvilinear protofibrils taken from the end of the lag-phase of Aβ$_{40}$ and Aβ$_{42}$ assembly (starting from 10 μM of peptide monomers), produced marked disruptions in the membrane surface, causing discontinuity

| Seeds / Monomer | Aβ$_{40}$ | Aβ$_{42}$ | p3$_{40}$ | p3$_{42}$ |
|---|---|---|---|---|
| Aβ$_{40}$ | **100%** | **-9%** | **76%** | **12%** |
| Aβ$_{42}$ | **-6%** | **100%** | **1%** | **86%** |
| p3$_{40}$ | **78%** | **-4%** | **100%** | **18%** |
| p3$_{42}$ | **9%** | **72%** | **20%** | **100%** |

**Fig. 5 | Tabulated $t_{50}$ for cross-seeding.** The $t_{50}$ values for each combination of monomer with different seeds, presented as a percentage relative to self-seeding $t_{50}$ values. Values are from the data shown in Fig. 4. and Supplementary Fig. 8 and Table 2. The values highlight profound seeding >70% for seeding between the p3$_{40}$:Aβ$_{40}$ and p3$_{42}$:Aβ$_{42}$ pairs, shown in green. In contrast, other combinations of peptides had a negligible impact on lag-times, shown in red. The relative impact on $t_{50}$ values, expressed as a percentage, are calculated as follows: (Unseeded $t_{50}$ - Cross seeded $t_{50}$ / Unseeded $t_{50}$ - Self seeded $t_{50}$) x 100%.

and distortions in the membrane, Fig. 6b,c and Supplementary Fig. 9b, c. Inspection of >300 individual vesicles indicate >80% of the lipid membrane surface are disrupted in this way, Fig. 6f and Supplementary Table 3. We have previously shown Aβ$_{40}$ and Aβ$_{42}$ fibrils, or monomers, have little impact in lipid membrane integrity[36]. Here we

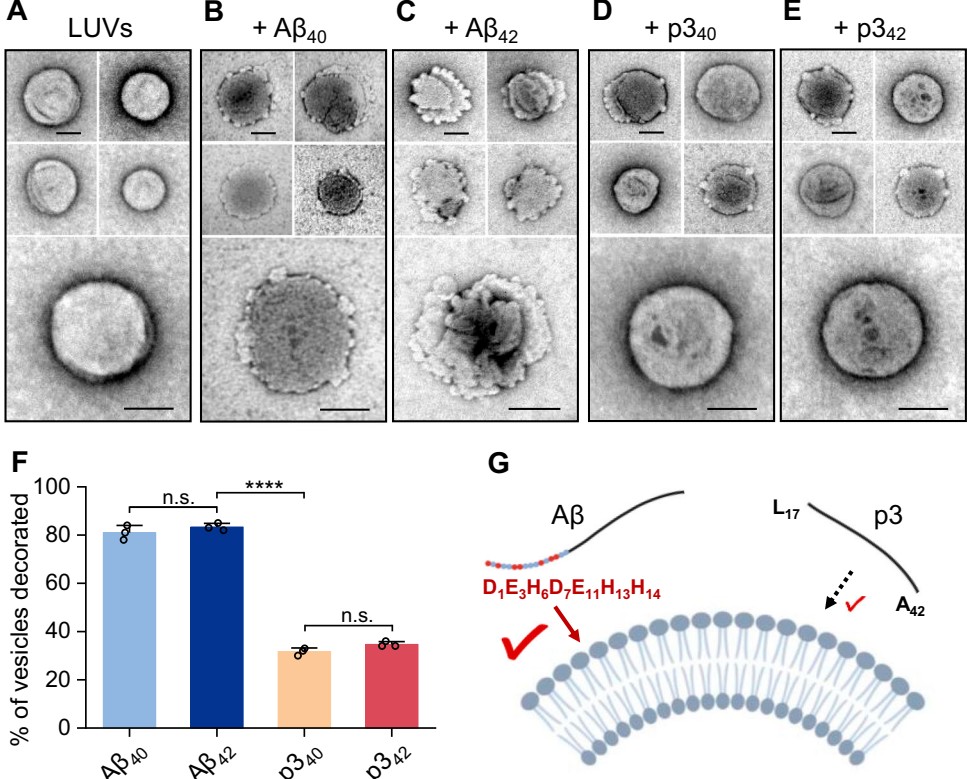

**Fig. 6 | Impact of Aβ₄₀, Aβ₄₂, p3₄₀ and p3₄₂ oligomers on lipid vesicles.**
**A–E** Large unilamellar vesicles (LUVs) in the absence of Aβ (**A**); incubated with: (**B**) Aβ₄₀ oligomers, (**C**) Aβ₄₂ oligomers, (**D**) p3₄₀ oligomers and (**E**) p3₄₂ oligomers. Aβ and p3 peptide oligomers are taken from the end of the lag-phase after incubation of 10 μM monomeric preparations. Scale bar: 100 nm. More examples in Supplementary Fig. 9,10. **F** Analysis of lipid vesicles impact of Aβ oligomers. Error bars are

SEM from three independent replicates, 100 vesicles imaged per replicate. One-way ANOVA test, the symbols n.s. and ****, indicate *P*-values of Aβ₄₀-Aβ₄₂, *P* = 0.49; p3₄₀-p3₄₂, *P* = 0.30; and ****P ≤ 0.0001, p3₄₂-Aβ₄₂, respectively. **G** Cartoon highlighting loss of charged residues causes a reduced interaction with the phospholipid bilayer.

show the p3 peptides, p3₄₀ and p3₄₂ oligomers, also disrupt lipid vesicles, but this is less widespread, Fig. 6d, e. Inspection of >300 vesicles suggests 32% and 35% of vesicles are disrupted, Fig. 6d-f, Supplementary Fig. 10 and Supplementary Table 3. This data implies a reduction in the impact p3 oligomers have on the lipid bilayers.

Next, we wanted to investigate to what extent the presence of the p3 peptides can disrupt the cellular membrane and allow Ca²⁺ and other ions to flow into the cell lumen. For this study we chose HEK293 cells, which have neuronal character, and match the cell-line used for the well-established patch-clamp studies described later. Using a Ca²⁺ sensitive fluorescent dye, Fluro-3, we have monitored the impact of Aβ and p3 peptides on cellular membrane permeability. In agreement with the TEM images, Aβ₄₀ and Aβ₄₂ oligomers cause influx of Ca²⁺ into the cell, within 30 s after addition of Aβ, for almost 100% of the preparations studied, Fig. 7a, b. Previous studies have shown Aβ₄₂ induced Ca²⁺ influx is from an extra-cellular source, not from intra-cellular stores[40]. The p3 oligomers cause cellular Ca²⁺ influx less frequently, with 54% and 64% of cell preparations having detectable Ca²⁺ influx, for p3₄₀ and for p3₄₂, respectively, Fig. 7c-e. Furthermore, for preparations that do exhibit Ca²⁺ influx, the fluorescence intensity is also reduced, Fig.7f. The HEK293 cells have less Ca²⁺ influx when exposed to p3 oligomers relative to Aβ. This supports the assertion that the N-terminal residues, although not essential for lipid disruption, do enhance the level of interaction with the lipid bilayer.

Cell viability measurements support the differential effects of Aβ and p3 on membrane permeability, Fig. 8. We studied both rat primary neurons, Fig. 8a, and HEK293 cells, Fig. 8b, c. Assessment of cell viability, measured using an MTT assay, suggests that Aβ₄₂ oligomers are the most cytotoxic form, followed by Aβ₄₀. The p3₄₂/₄₀ oligomers were

also cytotoxic but less so than Aβ₄₂ oligomers, Fig. 8a, b. Cell treatment with fibril preparations of Aβ and p3 result in less cytotoxicity, with only a slight reduction in cell viability relative to treatment with the buffer control, Fig. 8c.

Aβ contains most of its charged residues in the N-terminus, residues 1–16, with four carboxylate sidechains as well as three charged histidine side-chains. The loss of these in the p3 peptide will mean less interaction with charged phospholipid head-groups. An initial electrostatic interaction with the residues in the N-terminus of Aβ, followed by insertion into the membrane, is suggested as a mechanism for Aβ interactions with lipid membranes[60,61].

## p3 annular ring assemblies
It is postulated that Aβ cytotoxic action is caused by its ability to form annular structures, that insert into the membrane and form large unregulated ion-channel pores[41-44]. We wondered if p3 oligomers could form these ion-channels, and what impact the loss of the N-terminal residues (carboxylates and histidine's) would have on the ion-channel properties.

Analysis of TEM images of p3₄₀ and p3₄₂ peptides, taken at the end of the lag-phase, reveal a snapshot of prefibrillar assemblies. Curvilinear protofibrils, but also many oligomeric structures that resemble annular ring-like assemblies, are widespread in the micrographs. Typical micrographs are shown in Supplementary Fig. 5. The p3 annular structures, shown in Fig. 9, have a very similar appearance to those reported for Aβ₄₂, which have recently studied by cryo-EM and cryo-ET[45]. We have used single particle analysis to generate 2D class averages for both p3₄₀ and p3₄₂ oligomers, Fig. 9. Particularly noteworthy are the class averages, which have a ring-shaped

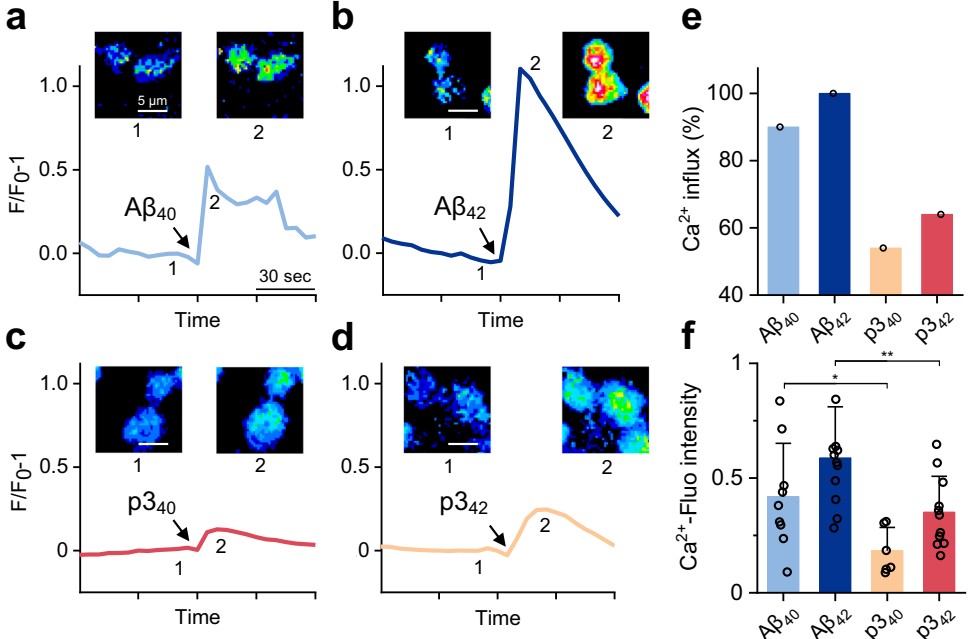

**Fig. 7 | Calcium influx of HEK293T cells in response to different Aβ and p3 oligomer preparations. a–d** Time-lapse recording of fluorescence relative to fluorescence before addition of (**a**) $A\beta_{40}$ oligomer, (**b**) $A\beta_{42}$ oligomer, (**c**) $p3_{40}$ oligomer and (**d**) $p3_{42}$ oligomer. Also shown are single-cell florescence images before (labeled 1) and after (labeled 2) the addition of Aβ or p3 oligomer.

**e** Percentage of preparations showing $Ca^{2+}$ influx. **f** Mean $Ca^{2+}$ fluorescence intensity, $(F/F_0)$-1, where F is the observed fluorescence and $F_0$ is fluorescence just before addition of Aβ peptides. Error bars are SEM from six replicates. Two tailed pair-wise $t$-test, the symbols * and **, indicate $P$-values of 0.038 and 0.008, respectively.

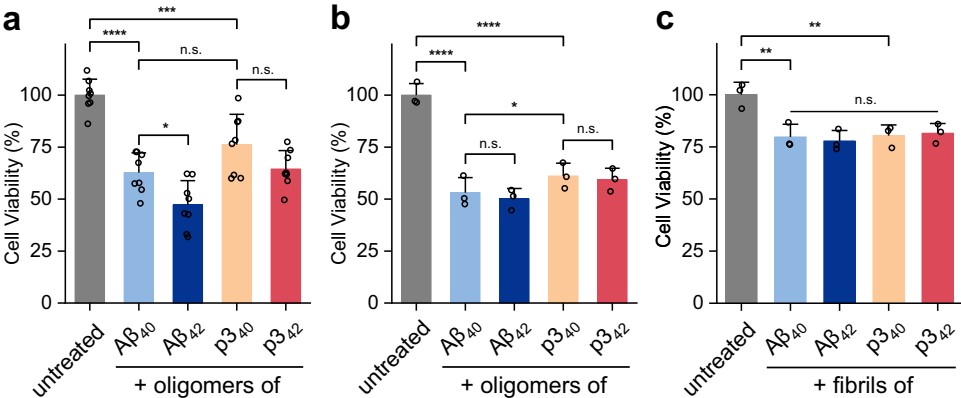

**Fig. 8 | Cell viability after treatment with Aβ or p3. a** Cytotoxicity of oligomers on rat primary neurons. **b-c** Cytotoxicity of oligomers (**b**) and fibrils (**c**) on HEK293 cells. The cells were incubated for 24 h with oligomers or fibrils. $A\beta_{40}$ (pale blue), $A\beta_{42}$ (dark blue), $p3_{40}$ (orange) and $p3_{42}$ (red). Replicate measurements were $n = 8$ and $n = 3$ for primary neurons and HEK293 cells respectively. Error bars indicate standard error of the mean. One-way ANOVA test (normal distribution confirmed with Shapiro-Wilk test), the symbols n.s., *, **,*** and ****, indicate $P$-values of (**a**)

$A\beta_{40}$-$p3_{40}$ $P = 0.11$; $p3_{40}$-$p3_{42}$ $P = 0.21$; *$P = 0.0484$; ***$P = 0.0008$ and ****$P \leq 0.0001$ (**b**) $A\beta_{40}$-$A\beta_{42}$ $P = 0.97$; $p3_{40}$-$p3_{42}$ $P = 0.99$; *$P = 0.50$ and ****$P \leq 0.0001$; (**c**) control-$A\beta_{40}$ **$P = 0.007$ and control-$p3_{40}$ **$P = 0.008$. Aβ and p3 preparations were from 10 µM monomer equivalent, the controls were cells treated with an equal amount of buffer. Cell viability, monitored by MTT essay, indicates that Aβ and p3 oligomers were cytotoxic to rat primary neurons and HEK293 cells.

appearance. Dimensions of the external ring are *ca.* 7–10 nm with an internal channel ~1–2 nm in diameter. These annular assemblies of p3 have not previously been reported and add to the number of amyloid forming proteins that also form annular structures[62]. The curvilinear protofibrils, also apparent, have a consistent width. These have been measured to be 2.8 nm, recently characterised for $A\beta_{42}$ by cryo-EM and cryo-ET[45]. Despite the loss of residues 1–16, the p3 peptides are capable of forming curvilinear protofibrils and annular structures that are very similar to those reported for full-length Aβ[45]. Residues 1 to 14 have been shown to be very flexible in fibrils, with a lack of stable hydrogen-bonds in this region[63]. These N-terminal residues may therefore not contribute to the stability of fibrils or pre-fibrillar assemblies.

## Ion channel pore formation

To study the ion-channel forming properties of the p3 peptide, we used excised membrane patches from HEK293 cells and performed patch-clamp electro-physiology measurements. Here we have used an 'inside-out' configuration to allow p3 peptides to diffuse to the extracellular surface of HEK cell membranes. We then measured channel conductance across the lipid-bilayer using a stepped voltage of -80, 0 and +80 mV, typical examples of which are shown in Fig. 10 and Supplementary Fig. 11. Large conductances, of *ca.* 200 pS for $p3_{42}$ peptide are observed, these have a flickering appearance, rapidly opening and closing, Fig. 10a, b. They are present in both, polarising and depolarising potentials. The ability of the channels to almost instantly fully

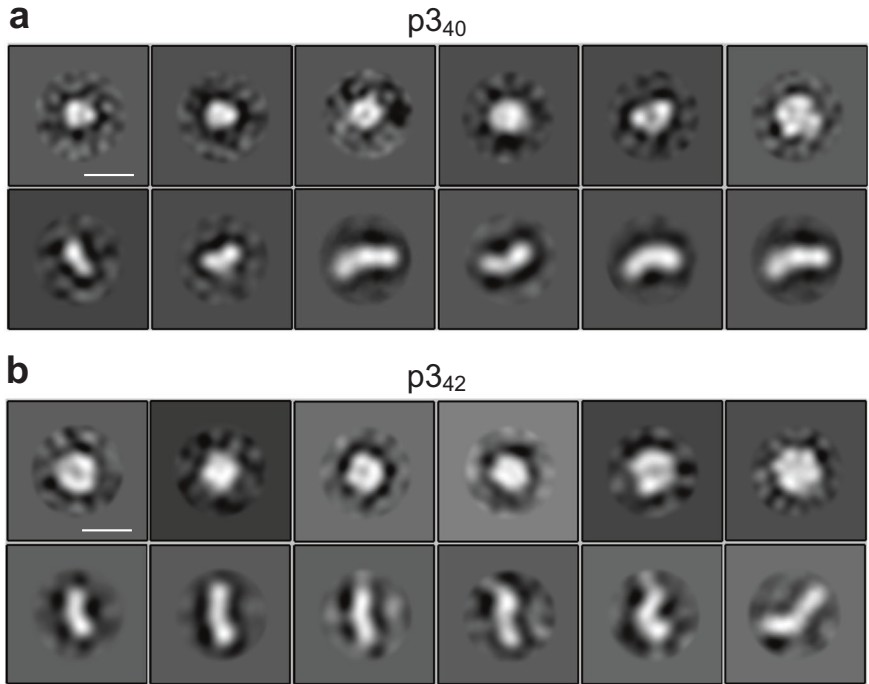

**Fig. 9 | 2D class averages of oligomers and curvilinear protofibrils for (a) p3$_{40}$ and (b) p3$_{42}$.** *Ca.* 150 single particles for each of the 2D class averages. Top rows show annular ring structures for both p3$_{40}$ and p3$_{42}$. While bottom rows show curvilinear protofibrils with consistent widths. Oligomers are negatively stained with uranyl acetate. Scale bar: 10 nm.

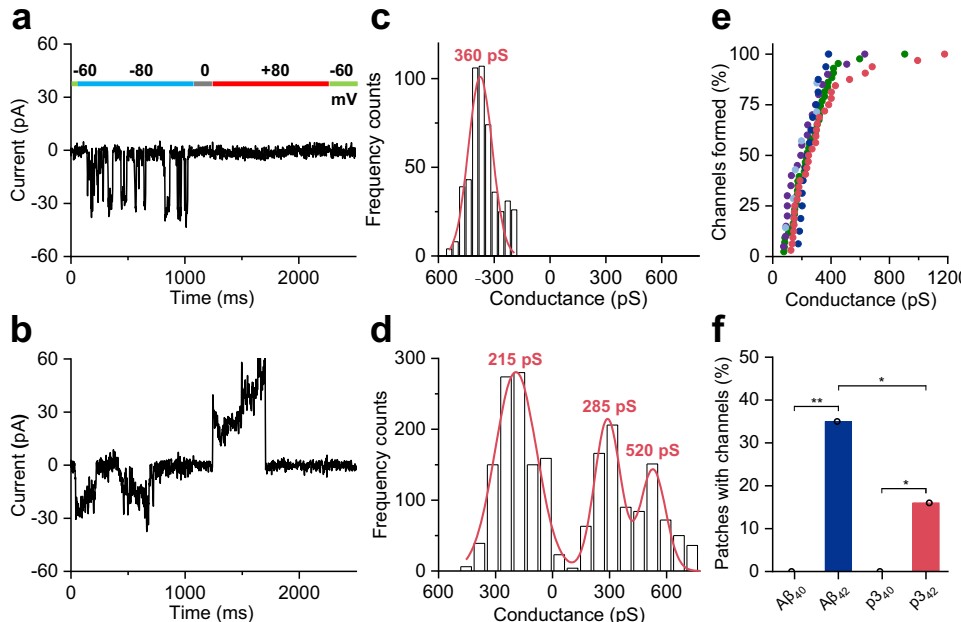

**Fig. 10 | Ion channels of p3$_{42}$ oligomers. a, b** Patch-clamp current recordings, 2.5 s, across HEK293 cell membrane patches, with extra-cellular p3$_{42}$ oligomers (5 µM). **c, d** Frequency distribution of conductance for the same traces, gaussian curves indicate conductance values. On the left conductance values when a negative potential was applied (-80 mV), whereas on right conductance value are when the applied potential was positive (+80 mV). **e** Range of conductance observed for p3$_{42}$ for five individual membrane patches. **f** Percentage of membrane patches that record ion channels for Aβ and p3 oligomers. Fisher test, the statistical test used was two-sided, the symbols * and ** indicate *P*-values of Aβ$_{40}$-Aβ$_{42}$ **$P = 3 \times 10^{-7}$; p3$_{40}$-p3$_{42}$ *$P = 0.05$; Aβ$_{42}$-p3$_{40}$ *$P = 0.06$.

close and open, suggests these conductance traces are typically for single channels. The size of the conductances can be related to channel pore-size by p3 annular oligomers. A 2,500 ms measurement (in pA) are converted to conductance (pS) and plotted as a histogram to reveal the range of conductance values, Fig. 10c, d. These have been fitted to a gaussian curve so as to obtain a typical conductance value for each 2500 ms recording.

The range of conductances is quite variable, Fig. 10e, typically between 150–355 pS (20–80% of the total range). This range of conductance values are similar to Aβ$_{42}$, previously reported,[44,45] with

conductance values ranging between 200–575 pS, Supplementary Fig. 12. The loss of residues 1–16, including a number of charged residues, does not markedly affect the conductance properties of the channel pores, although median conductance values are a little smaller, by 140 pS. Assuming a channel length of 54 Å (the lipid bilayer thickness) these conductance values suggest a channel diameter of 1.1 to 1.5 nm[44,45]. This is in-line with the internal diameter of the annular structures shown in Fig. 9 for $p3_{42}$.

$A\beta_{42}$ oligomers produce channels in excise cellular membranes for 35% of patch-clamp recordings[44]. The numbers of channel conductance observed for $p3_{42}$ is less, five pulled patches produced channels from 32 recordings (16%), Supplementary Table 4. The reduction in the number of channels formed, 16% compared to 35% for full-length is significant ($P < 0.1$, Fisher test). The reduced interaction of $p3_{42}$ with the membrane is also suggested from the vesicle imaging, Fig. 6, and $Ca^{2+}$ influx measurements, Fig. 7. The loss of sixteen N-terminal residues reduces the interaction with the membrane.

Importantly, we show only $p3_{42}$ peptide inserts into cellular membranes and forms channels but not $p3_{40}$ (Fig. 10, Supplementary Fig.11, Supplementary Table 4). This mimics the behaviour of full-length $A\beta_{40}$ and $A\beta_{42}$, where only $A\beta_{42}$ forms a large conductance across the membrane, Fig. 10f. This significant distinction in channel forming properties of $A\beta_{40}$ and $A\beta_{42}$ is important, as it parallels the pathology of the two peptides; $A\beta_{42}$ is thought to be the pathogenic isoform while $A\beta_{40}$ may be harmless[64–66]. The channel forming ability of $p3_{42}$, but not $p3_{40}$, supports the assertion that the $A\beta_{42}$ peptides pathogenicity is link to its ion-channel forming ability[44].

The membrane conductance recorded appear with a median time of *ca.* 5 min after exposure to $p3_{42}$ oligomers. Presumably it takes this time for the annular oligomers to insert and rearrange within the membrane. Full-length $A\beta_{42}$ also takes a number of minutes for the conductance to appear, with a median time to produce channels of 8 min[44]. The time for the ion channels to appear is distinct from the more general membrane permeability, measured by $Ca^{2+}$ influx, Fig. 7, which occurs almost instantly after exposure to $A\beta_{40/42}$ or $p3_{40/42}$ peptides.

To conclude, as previously highlighted by Raskatov, the p3 peptides have been misnamed and are in fact highly amyloidogenic[21,22]. We suggest that the cleavage pathway of Aβ and p3 should be referred to as the β- and α-secretase pathways, rather than amyloidogenic and non-amyloidogenic, to avoid confusion, Fig. 1. Our cross-seeding studies suggest rapid fibril formation of both $p3_{40}$ and $p3_{42}$ can trigger Aβ oligomer formation in vitro. The cross-seeding behaviour is highly specific and occurs only when the C-terminal lengths are matched. With the presence of the p3 peptide identified in diffuse amyloid deposits from AD brains, it is conceivable this cross-seeding behaviour may occur in vivo, further studies, using brain-derived p3 fibrils, will be needed in the future. The $p3_{40/42}$ peptides are capable of forming cytotoxic oligomers, curvilinear-protofibrils and annular structures. Like Aβ, these pre-fibrillar assemblies can cause membrane disruption, permeability and ion-channel pores in their own right. A role in AD pathology for this under-studied, hydrophobic, N-terminally truncated form of Aβ should certainly be considered.

## Methods
### Ethical Statement
The cultures were produced using tissue from Sprague Dawley rat pups. Dams (strain code: CD(SD)) were obtained from the breeder (Charles River, UK), and maintained and humanely culled in accord with the UK Animals (Scientific Procedures) Act 1986 Amendment Regulations 2012 (SI 2012/3039) and the EU Directive 2010/63/E.

### Aβ and p3 peptides
All synthetic Aβ peptides were purchased from EZBiolab Inc in a lyophilized form. Aβ peptides were synthesized by solid-phase F-moc (N-(9-fluorenyl)methoxycarbonyl) chemistry, and were purified with reverse-phase high performance liquid chromatography, Supplementary Fig. 13. Sequences was verified by electrospray ionization mass spectrometry (ESI-MS), Supplementary Fig. 13. ESI-MS measurements were performed on Agilent 6125 LC/MSD mass spectrometer (Agilent Technologies, Santa Clara, CA, USA). Peptides were desalted using a desalting column and samples were directly injected. Data were recorded in positive mode in an m/z range of 200 – 2000. Data were processed using MSD Chemstation software (Agilent Technologies). The following amino acid sequences were generated:

Full-length $A\beta_{40}$: DAEFR HDSGY EVHHQ KLVFF AEDVG SNKGA IIGLM VGGVV

Full-length $A\beta_{42}$: DAEFR HDSGY EVHHQ KLVFF AEDVG SNKGA IIGLM VGGVV IA

$p3_{40}$ Peptide $A\beta_{17-40}$: LVFF AEDVG SNKGA IIGLM VGGVV

$p3_{42}$ Peptide $A\beta_{17-42}$: LVFF AEDVG SNKGA IIGLM VGGVV IA

Unless otherwise stated, all other chemicals were purchased from Sigma-Aldrich and ThermoFisher scientific.

### Aβ and p3 solubilization and isolation
The lyophilized full-length Aβ and p3 peptides were solubilized in ultra-high quality water to 0.7 mg ml$^{-1}$ and adjusted to pH 10 with NaOH and left at 4 °C for 30 min. Then, the peptide solutions were centrifuged for 15 min at 20,000 g at 4 °C, to remove any high molecular weight aggregates. Crucially, size-exclusion chromatography (SEC) was used to remove any remaining nucleating and oligomeric aggregates for generating a seed-free preparation described as monomeric. Prefibrillar samples of Aβ and p3 peptides described as oligomeric were obtained as a heterogeneous mixture from the end of the lag-phase, as monitored by ThT in adjacent wells. These oligomeric preparations contain a range of curvilinear protofibrils and annular assemblies as well as an appreciable number of peptide monomers. Fibril samples were taken at the plateau-phase of fibril formation.

### Size exclusion chromatography (SEC)
Monomeric Aβ was isolated using AKTA FPLC with a Superdex 75 10/300 GL column (volume = 24 ml; GE Healthcare) at a flow rate of 0.5 mL·min$^{-1}$ at 4 °C. The column was pre-equilibrated with 30 mM sodium phosphate buffer (pH 7.4). The wild-type $A\beta_{40}$ and $A\beta_{42}$ peptide concentrations were determined using the single tyrosine absorption at 280 nm, $\varepsilon_{280} = 1280 \cdot cm^{-1}\ mol^{-1}$. $p3_{40}$ and $p3_{42}$ sequences lack a tyrosine residue (Tyr10), so peptide concentrations were determined by the amide absorbance at 205 nm ($\varepsilon_{205} = 83370\ M^{-1}\ cm^{-1}$ for $p3_{40}$ and $\varepsilon_{205} = 88930\ M^{-1}\ cm^{-1}$ for $p3_{42}$) using an online protein calculator[67]. Negative-stain electron microscopy and thioflavin-T fluorescence assay confirmed that SEC-purified peptides were seed-free. Monomeric peptides were used directly after SEC, Supplementary Fig. 14.

### Fibril growth assay
The kinetics of amyloid fibril formation were monitored with thioflavin T (ThT), a fibril-specific fluorescent dye[68]. Monomeric Aβ or p3 peptides (10 µM) with ThT (20 µM) were placed in a 96-well plate (200 µL per well) in 30 mM sodium phosphate buffer (pH 7.4) at 30 °C. The 96 well-plates were clear flat-bottom polystyrene, non-pyrogenic, non-tissue-culture treated plates (Corning 3881, USA). The well-plate remained quiescent throughout. ThT fluorescence was recorded by a FLUOstar Omega microplate reader (BMG Labtech, Aylesbury, UK) using the FLUOstar Omega software package, with excitation and emission filters at 440 nm and 490 nm respectively. Fluorescence readings were taken every 30 min. In the seeded aggregation assay, fibrils seeds (10%, 1 µM, Aβ or p3) were obtained by incubating 10 µM Aβ peptides in 30 mM sodium phosphate buffer at 30 °C for 3 days. Samples also contained DMSO 0.5% (0.5 mL/100 mL) used to solubilize ThT. The formation of Aβ and p3 fibrils was confirmed by ThT fluorescent assay and TEM imaging.

## Fitting fibril growth curves

The empirical kinetic values for $t_{lag}$ (typically the time required for the ThT fluorescence to reach 10% of the maximum value) and, $t_{50}$ (time required for half maximal fluorescence reached) were extracted from the data by fitting the fibril growth curve to the following equation[69].

$$Y = (y_i + m_i x) + \frac{v_f + m_f x}{1 + e^{-(\frac{x - xo}{\tau})}} \quad (1)$$

Where Y represents the ThT fluorescence intensity, and x represents the time. $x_o$ represents the time at which half maximal ThT fluorescence intensity is reached, referred to as $t_{50}$. The lag-time ($t_{lag}$) is taken from $t_{lag} = X_0 \cdot 2\tau$. The initial and final fluorescence signals is represented by $y_i$ and $\gamma_f$, respectively[69].

## Analysis of p3 peptide aggregation kinetics with AmyloFit

Global kinetic analysis of amyloid formation was preformed using the AmyloFit platform[48]. The amyloid fibril formation traces are described by the following integrated rate law, based on Michaelis-Menten-like kinetics:

$$\frac{M}{M(\infty)} = 1 - \left(1 - \frac{M(0)}{M(\infty)}\right) e^{-k_\infty t} \times \left(\frac{B_- + C_+ e^{\kappa t}}{B_+ + C_+ e^{\kappa t}} \times \frac{B_+ + C_+}{B_- + C_+}\right)^{\frac{k_\infty^2}{k_\infty \kappa}} \quad (2)$$

where the additional coefficients are functions of $\kappa$ and $\lambda$:

$$C_\pm = \frac{k_+ [P]_0}{\kappa} \pm \frac{k_+ M(0)}{2m(0)k_+} \pm \frac{\lambda^2}{2\kappa^2} \quad (3)$$

$$k_\infty = 2k_+ P(\infty) \quad (4)$$

$$\bar{k}_\infty = \sqrt{k_\infty^2 - 2C_+ C_- \kappa^2} \quad (5)$$

$$B_\pm = (k_\infty \pm \bar{k}_\infty)/2\kappa \quad (6)$$

which are two combinations of the microscopic rate constants of:

$$\lambda = \sqrt{2k_+ k_n m(0)^{nc}} \quad (7)$$

$$\kappa = \sqrt{2m(0)k_+ \frac{m(0)^{n2} k_2}{1 + m(0)^{n2}/K_M}} \quad (8)$$

where $m(0)$ represents the initial monomer concentration, $M(0)$ represents initial fibril mass concentration, $M(\infty)$ represents mass concentration of fibrils at equilibrium, $P(0)$ represents the initial aggregate concentration and $P(\infty)$ represents the aggregates concentration at equilibrium. The microscopic rate constants for primary nucleation ($k_n$), secondary nucleation ($k_2$), and elongation rate ($k_+$). The exponents $n_c$ and $n_2$ are the reaction orders for primary and secondary nucleation respectively. $K_M$ is the saturation constant for secondary nucleation. $n_c$ and $n_2$ were set as global constant of 2, so as not over-fit the data.

Wild-type $A\beta_{40}$ and $A\beta_{42}$ fibril assembly was fitted to a secondary nucleation model. The experimental macro kinetic traces were globally fitting to the integrated rate law over the range of p3 peptide concentrations. The microscopic rate constants $k_+ k_n$; $k_+ k_2$; and $k_+$ values were fitted to the fibril growth curves at p3 peptide concentrations of 12.5 μM, the other kinetic traces at decreasing concentrations, were then fitted in three scenarios in which only one of the rate constants were permitted to vary, while the other two remain constant. This approach has been used to investigate how increasing concentrations of an inhibitor of fibril formation effect individual microrate constants[70].

## Circular dichroism spectroscopy

Circular dichroism (CD) experiments were performed on an Applied Photophysics Chirascan instrument. CD spectra were obtained using a cuvette with a 1 mm pathlength, at 25 °C, under a nitrogen atmosphere. Spectra were recorded in the range of 190 nm – 260 nm with sampling points every 0.5 nm. Each spectrum represents the average of three scans. Data were processed using Applied Photophysics Chirascan Viewer and Origin software. The molar ellipticity [θ] (deg.cm$^2$.dmol$^{-1}$) were calculated using the equation [θ] = mdeg/ (l c), where 'l' is the pathlength and 'c' is the molar concentration.

## Transmission electron microscopy

Aβ and p3 peptide samples were generated with the same protocol used in the fibril growth assay, but without ThT addition. Oligomer samples were taken at the end of the lag-phase during fibril formation, while fibril samples were taken at the equilibrium phase. Aliquots (5 μL) of peptide samples (10 μM peptide) were added to carbon-coated 300-mesh copper grids, after being glow discharged (Agar Scientific Ltd, Essex, UK). The grids were then blotted after 90 s and rinsed with ddH$_2$0 at room temperature. Uranyl acetate (5 μL of 2 mg/100 ml) was added, then blotted and rinsed after 10 s. Glow discharge was carried out using the Pelco EasiGlow unit. Images were recorded by a JEOL JEM-1230 electron microscope (JEOL Ltd., Japan) at 40,000 magnifications, operated at 120 kV, paired with a 2k Morada CCD camera and corresponding Olympus iTEM software package (Olympus Europa, UK). Crossover distances and fibril widths were measured by the segmented-line mode of the image-J software.

## Single particle analysis

Single particles were manually picked from 100 negatively stained TEM micrographs. 2D class-averages, for each preparation were generated, using RELION-4.1 software. CTFFIND-4.1 program was used to estimate the contrast transfer functions (CTFs) of the images. Approximately 2500 particles were manually picked from each of the datasets and the particles were extracted with a 31 nm box size. This was followed by a 2D classification into 20 classes, employing a mask with diameter of 200 Å. Representative 2D classes of annular oligomers and curvilinear protofibrils are shown in Fig. 9.

## Vesicle preparation

Large unilamellar vesicles (LUVs) were prepared using an extrusion method described previously[36]. The lipids used were egg phosphatidylcholine (PC) dissolved in chloroform, cholesterol dissolved in chloroform and the mono-sialotetrahexosl-ganglioside (GM1) dissolved in ethanol, at a ratio of 68:30:2 (by weight). Lipid solutions were placed in a fume hood overnight, to allow the organic solvent to evaporate. The lipid film was then re-solubilized in 30 mM sodium phosphate aqueous buffer (pH 7.4), to a lipid concentration of 1 mg ml$^{-1}$. LUVs were generated using a benchtop mini extruder (Avanti Polar Lipids, Alabama, USA) at 60 °C. The lipid solution was passed across a 100 nm polycarbonate membrane for 21 times. Lipids were stored at 4 °C for a maximum of 2 days before use.

## Cellular Ca(II) influx

Fluo3-AM loaded HEK293T cells: HEK293T cells were incubated at 37 °C, in 5% CO$_2$ containing Dulbecco's modified Eagle's medium (DMEM, purchased from Thermofisher) supplemented with 10% foetal bovine serum and 0.2 mg ml$^{-1}$ penicillin-streptomycin. Cells were plated into a 12-well plate (1 mL each well) incubated overnight, and cells were grown to 70–80% confluence. Next, cell media was replaced with fresh DMEM containing 5 μM Fluo3-AM (Abcam). To allow the cellular uptake of the Ca$^{2+}$ sensitive fluorescent dye, plates were left in an

incubator for a further 30 min. The cells were then washed twice with 400 μL of DMEM Eagles cell-media in each well, to remove the extracellular Fluo3-AM. Cells were incubated for a further 20 min at 37 °C, to allow de-esterification of intracellular AM esters, which activates intracellular $Ca^{2+}$ dependent fluorescence. Finally, the DMEM was replaced with an aqueous buffer containing $CaCl_2$ (1.8 mM); NaCl (120 mM); CsCl (10 mM); HEPES (9 mM); KCl (2.2 mM); and $MgCl_2$ (1.9 mM), buffered to pH 7.4. The Fluo-3 loaded cells will then be ready for time-lapse fluorescence microscopy imaging.

$Ca^{2+}$ Fluorescence Imaging of Fluo3: Fluorescence microscopy was performed using an inverted Leica DM IL microscope at 10x magnification. The bandpass filter allowed excitation at 470 nm, and emission was recorded at 520 nm. A charge-coupled device (CCD) camera was used to acquire time-lapse fluorescence images and bright-field visible light images, with a temporal resolution of one image every 5 s, recordings were for typically 9 min.

Aβ (Aβ$_{40}$ oligomers; Aβ$_{42}$ oligomers) and truncated Aβ peptide-p3 (Aβ$_{17-40}$ oligomers; Aβ$_{17-42}$ oligomers) were studied. Peptides samples with prefibrillar oligomers and curvilinear protofibrils were taken during fibril assembly at the end of the lag-phase. The 30 μM Aβ stock solutions was added to HEK293 cells within 400 μL buffer to produce 5 μM final Aβ concentration. Images were acquired using the ProgRes CapturePro 2.8.8 software and fluorescence intensity was measured by Image-J software. The time series analyser, V3 plug-in, was used to measure fluorescence intensities by analysing the overall field. We plotted $(F/F_0)$-1 as the change in Fluo-3 fluorescence signals with time, where (F) represents the observed fluorescence and ($F_0$) represents the background fluorescence at a time point just before the addition of Aβ or p3 peptides. Typically, $N = 10$–14 preparations with three independent repeats were obtained. Addition of Ionomycin, which is known to cause pronounced Ca(II) influx, and was used as a positive control. Ionomycin extra-cellular concentration was 5 μM and induced a $Ca^{2+}$ fluorescence median signal of 1.8, $(F/F_0)$-1.

### Cytotoxicity assay
Rat primary neuronal cells and HEK293T cells: Cortical tissue was dissected from a Sprague Dawley rat embryo brain obtained at embryonic day 18, and the dissociated cells were seeded onto the 60 inner wells of three poly-D-lysine-coated (10 μg/mL) 96-well plates, at 20,000 cells/cm², in 100 μl of modified Neurobasal medium (Invitrogen, UK) per well. The plates were left undisturbed in a cell culture incubator at 37 °C and 5% $CO_2$ for 7 days before peptide application, half of the culture medium being replaced every 2 days. HEK293T cells were cultured in a 96-well plate for 24 h at 37 °C and 5% $CO_2$ in DMEM medium (Gibco) containing 10% Foetal Bovine Serum (Gibco). HEK293T cells were cultivated in a 96-well plate for 24 h.

Aβ and p3 peptide application: Aβ and p3 oligomers were obtained at the end of the lag-phase, with ThT in adjacent wells, while fibrils were obtained at the end of fibril assembly at equilibrium. The p3 and Aβ preparations were applied at a final concentration of 10 μM by removing half the volume of medium in each well and replacing it with an equivalent volume of p3$_{42;}$ p3$_{40;}$ Aβ$_{42}$ or Aβ$_{40}$ peptide preparations. HEPES buffer (20 mM, pH 7.4) was added to the wells that served as the controls. The cells were left undisturbed for 24 h post-treatment before their viability was assessed.

Cell viability assay: Cell viability was measured by the MTT assay. The MTT reagent (10 μL; 5 mg mL⁻¹ purchased from Merck) was added to each well and left to incubate for 4 h at 37 °C. Isopropanol with 0.04 M HCl (10 μL) was added to each well. The concentration of MTT formazan produced by live cells was measured using the absorbance at 570 nm in a microplate reader (FLUOstar Omega, BMG LABTECH) within 1 h. Cell viability was presented relative to that of cells with an amount of buffer added to the cells equal to that used for peptide treatments.

### Patch clamp electrophysiology
Cell culture: HEK293 immortal cells underwent incubation in a 30 ml cell culture flask, at 37 °C, with a 5% $CO_2$ environment in Dulbecco's Modified Eagle Medium, supplemented with 10% fetal bovine serum, and penicillin-streptomycin (0.2 mg ml⁻¹) mixture. Cells were subjected to division when reaching ~70–80% confluency, at intervals of ~5–7 days, employing $Ca^{2+}$ and $Mg^{2+}$free phosphate-buffered saline (pH 7.2), meanwhile a portion of the cells was seeded onto 35 mm diameter easy-grip Petri dishes with the same buffer. These cells were allowed to incubate for 2-3 days prior to patch-clamp recordings.

Patch clamp recordings: All patch clamp measurements were conducted using excised membrane in an 'inside-out' configuration. This type of configuration facilitating the exposure of the Aβ and p3 oligomers, to the extracellular surface of plasma membrane of the HEK293 cells, while under a voltage-clamp mode. Oligomers were obtained at the end of the lag-phase of fibril assembly. Tip burnt polished micro glass pipettes (GC150TF-10, Harvard Apparatus) were backfilled with extracellular buffer containing NaCl (120 mM); CsCl (10 mM); HEPES (9 mM); KCl (2.2 mM); and $MgCl_2$ (1.9 mM); $CaCl_2$ (1.8 mM) and EGTA (0.1 mM) with addition of 5 μM (monomer equivalent) of Aβ or p3 oligomers at pH 7.4. Membrane patches measuring ~1–3 μm in diameter were pulled, and Aβ and p3 oligomers were permitted to diffuse towards the extracellular membrane surface within the pipette. The pipette's resistance ranged from 2–6 MΩ when filled with the recording solution. Junction potentials manifested at the interfaces of ionic asymmetry and were duly accounted for the application of a pipette offset potential. Recordings were sampled at a frequency of 2 kHz with intervals of 500 μs, employing a lowpass 4-pole Bessel filter with a frequency cutoff of 1 kHz. The holding potential was established at -60 mV.

Channel-induced transmembrane currents were recorded by voltage clamping for 30–45 min. Employing the Axopatch 200B amplifier (Axon Instruments), a protocol ranging from -80 to 0 to 80 mV was implemented using pCLAMP 11 software. The processing of all recordings was conducted using Clampfit software, with the application of a lowpass boxcar filter.

### Reporting summary
Further information on research design is available in the Nature Portfolio Reporting Summary linked to this article.

## Data availability
All data supporting the results of this study can be found in the article, supplementary, and source data files. Source data are provided with this paper. Mass spectrometry data has been deposited in Figshare [https://doi.org/10.6084/m9.figshare.28343099].

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

## Acknowledgements

We are thankful for the support of the Biotechnology and Biological Sciences Research Council (BBSRC); project grant code BB/Y001931/1 and BB/M023877/1, awarded to J.V.; Chinese Scholarship Council (CSC), awarded to Y.T.; and Consejo Nacional de Ciencia y Tecnología (Conahcyt), awarded to A.T.

## Author contributions

Y.T. and J.V. designed the project. Y.T. and A.K. performed Aβ fibrillizations and kinetics. Y.T. and H.Z. conducted the TEM and data analysis. Q.S. and Y.T. performed Ca$^{2+}$ fluorescence imaging. Q.S., B.T., A.K., Y.T., and P.P. performed cytotoxicity assay. A.T and Y.T. performed patch clamp electrophysiology. Y.T., Q.S, A.T., P.P and J.V. wrote the manuscript, and all authors contributed to the correction and editing of the manuscript. J.V obtained funding and supervised the project.

## Competing interests

The authors declare no competing interests.
