## [Transparent Peer Review file · Nature Communications]

The p3 peptides (A β _{17-40/42}) rapidly form amyloid fibrils that cross-seed with full-length A β

Corresponding Author: Professor John Viles

Version 0:

Reviewer comments:

Reviewer #1

(Remarks to the Author)

In this manuscript the authors describe the characterization of the amyloidogenic potential of p3 fragments from APP. Historically, this has been considered to be the product of non-pathogenic (non-amyloidogenic) processing of APP. The authors report that the two major p3 peptides (40 and 42) can both efficiently form amyloid fibrils similar to full length A β peptides. These peptides also formed oligomers and annular structures like A β . Most importantly, the p3 peptides were able to seed the formation of A β peptide fibrils implicating them in Alzheimer's disease pathogenesis. Further characterizations included the association of p3 aggregates with membranes, membrane disruption (calcium leakage), cellular toxicity, and calcium pore formation. This very interesting work shows the potential importance of these abundantly produced p3 peptides to Alzheimer's disease pathogenesis as well as providing a plethora of important biochemical and physiological findings for them.

Major Concerns:

1) While these are exciting findings demonstrating many properties of p3 peptides and their effects on A β fibrillization, the clinical relevance is not yet fully determined. For this, the authors should provide direct evidence of p3 peptide fibrils isolated from human brain tissues. Most importantly, the demonstration of A β fibril seeding by p3 peptides derived from human brain tissue would go a long way to establishing the clinical relevance of this work.

2) Fibril formation by p3 (40) has been previously reported. As well, seeding of A β fibrillization has previously been reported (PMID: 32412731). It was also reported that p3 can assemble with A β and alter its biological properties (PMID: 38236059). Therefore, some of the data presented in the manuscript are not entirely novel and the core concept of p3 peptide fibrillization and seeding has already been reported.

Minor Concerns:

- i. Can thioflavin-T alter the fibrillization properties of peptides?
- ii. Within the abstract and some places in the body of the text, data is used in the singular tense however data are plural.

Reviewer #2

(Remarks to the Author)

Review of the manuscript entitled:

The p3 peptides, A β _{17-40/42}, rapidly form amyloid fibrils that cross-seed with 1 full-length A β , and the p342 annular oligomers form ion channel pores

The fundamental problem with the development of an effective therapy for Alzheimer's disease (AD) lies in the incomplete identification of the factor(s) responsible for its development. To date, our knowledge is based on longer, aggregation prone A β fragments (particularly A β 1-42 and 1-43) resulting from APP proteolytic processing and abnormally phosphorylated Tau protein. Additionally, it seems that only the oligomeric forms (not amyloid aggregates) of these peptides/proteins may play an important role in the development of the disease. This work focuses on the shorter A β fragments: A β ₁₇₋₄₀ and 17-42,

previously, perhaps prematurely, considered irrelevant in AD pathogenesis. The authors demonstrate that p3 peptides can form amyloid aggregates, cross-seed amyloidogenesis of compatible A β 1-40/42 counterparts and, more importantly, form cytotoxic oligomers (though slightly less toxic compared to A β 1-42/ 1-40). That is why the work seems to be important for exploring the mechanism of AD, albeit the comments mentioned below need to be addressed.

Key points:

1. In the Abstract (lines 21-23), the authors claim that: "The p3 peptide can cross-seed A β fibril formation, triggering oligomer and fibril formation in Alzheimer's disease, thus until now, their potential role in disease pathology has been largely misunderstood." while triggering oligomer formation by p3 peptide has not been directly demonstrated in the manuscript. I propose to rewrite this sentence in a more speculative form.
2. The authors analysed the effects of p3 peptides solely on HEK293 cells. Immortalised human embryonic kidney cells, although widely used in in vitro studies since the 1970s, are not an optimal model of neural cells. It would be very interesting to see, at least, the cytotoxicity of p3 peptides in the primary culture of neurons (or equivalent experimental model).
3. From the description in the Results & Discussion and Materials & Methods sections, it is not clear in what configuration the patch clamp experiments were actually conducted. On page 25 (lines 484-486): "All patch clamp measurements were conducted using excised membrane in an inside-out configuration, thereby facilitating the exposure of the A β oligomers, to the extracellular surface of plasma membrane HEK293 cells, while under a voltage-clamp mode." If the patch clamp experiments were in fact performed in an inside-out configuration, then it should be „intracellular surface of plasma membrane" instead of „extracellular surface of plasma membrane".
4. For how many TEM images are the presented in Fig 3 the p3 peptide oligomers representative? Please, if possible, show the original TEM images of ring-shaped p3 oligomers.

Minor points:

1. In Abstract (line 16), I suggest replacing "C-terminal lengths" with "C-terminal residues".
2. On page 7 (line 168) and page 19 (line 326), instead of "urinal acetate" it should be "uranyl acetate".
3. On page 10 (line 259) please replace "pre-fibular" with "pre-fibrillar".
4. TEM images in Fig. 3 should be presented at the same magnification, i.e., the oligomer images should be at the same magnification as the other images in the figure.
5. The same as above applies to Supplementary Fig. 5, i.e., to enable comparison of the morphological features of the amyloid aggregates, the image of A β 40 fibrils should be presented at the same magnification as the other images (with scale bar of 100 nm).
6. The citation style seems to be not uniform.

Reviewer #3

(Remarks to the Author)

This manuscript by Tian et al. details a biophysical study on the amyloid aggregation of A β 17-40/42 peptides. There are currently controversial and confusing evidence on these so called p3 peptides in terms of their amyloid forming potentials and biological implications in human diseases. This study incorporates a wide set of biophysical approaches to confirm that both of the A β 17-40/42 peptides do form amyloid fibrils at least in vitro and offer excellent quality data on the assembly mechanism and membrane disruption potential of the species formed during assembly. Therefore, I think this study presents a significant step forward regarding our molecular and mechanistic understanding on the amyloid assembly of the p3 peptides. Therefore, I am very supportive of this manuscript and recommend that it should be published after revisions. Below are some comments which I hope the authors will find helpful in revising and improving this manuscript.

- The key procedure that seems to have made a big difference compared to previous work and subsequently enabled all of the characterisation of the p3 amyloid assembly seems to be the additional SEC purification step. This is similar to historically how reproducibility for Abeta40/42 experiments was originally improved (e.g. Erik Hellstrand et al. 2010 <https://doi.org/10.1021/cn900015v>) Therefore, given the importance of this step, the result of this step should be better reported and example SEC trace should be presented in the supplementary information and compared to Abeta40/42 SEC experiments as this will help our community to reproduce and advance on experiments on p3 peptides .
- The evidence base that the formed structures are indeed cross-beta amyloid structures is already strong in this case. But for completeness I think the authors should still demonstrate the key tinctorial properties of amyloid in that the A β 17-40/42 structures they form in vitro also bind Congo red and produce birefringence
- The ThT kinetics data show in the manuscript are of excellent quality. However, given the known low reproducibility of these experiments, the authors should give more details of the variations observed. In the figure legends they report 3 replicates were recorded, this should be clarified as technical replicates. The batch-to-batch reproducibility of the series (e.g. on t50, gamma, etc) should be examined and commented. The exact type of plates used should be reported with product number details and the reading programme with the reding interval details should be given as all of these factors affect the reproducibility considerably.
- The authors reported the "node-to-node period" and the width of their fibril structures. How this was carried out is not currently described in the manuscript but should be reported. The "node-to-node period" the authors mention should be changed to cross-over distances, and I also suggest the authors to indicate both the cross-over distance and the helical pitch length reported in the cryo-EM structures of Abeta amyloid (e.g. in a SI table or visually in fig 3C) to help the readers to compare.
- The EM images and the distribution of cross-over distances and widths reported in fig 3 clearly indicate that the samples are highly polymorphic. The distributions are, in fact, showing that the differences between all four samples are substantial

and not “similar” as the authors state on page 5. The authors quote in the text mean value \pm standard deviation and this is very misleading as the casual readers will incorrectly infer similarity since the mean \pm standard deviation bounds overlap. However, in population analysis of polymorphic amyloid populations, these distributions are very different (e.g. one can have completely different distributions but very similar mean values, see Aubrey 2020 et al <https://doi.org/10.1038/s42004-020-00372-3>). Therefore, the authors should revise this section. The ranges of values should be quoted as ranges (e.g. the cross-over distances observed ranges from x to y). While the violin plots are visually nice, the distributions should also be plotted as normal histograms in the supplementary so that the readers can evaluate the differences and see the n-value used in the analysis. There should be also a short discussion or note on the clearly polymorphic nature of these samples in this section.

– The cross-seeding data are very interesting. I agree with the authors that structural difference in the seeds most likely is the reason for the pattern observed in fig 5 (and SI table 1 and SI fig 6). However, the caveat is that under the conditions employed by the authors, the dominating seeding mechanism seems to be surface nucleation rather than templated elongation because even at high concentration (10%) seeds the lag phase is still present (the ability by low amounts of seed to remove lag phase is one of the criteria used to confirm seeding by templated elongation, see Koloteva-Levine et al. 2021, <https://doi.org/10.1073/pnas.2104148118>). To clarify on this important mechanistic point, I suggest the authors either find and carry out the cross-seeding experiments under condition where self-seeding is dominated by templated growth, or to carry out seed monomer equivalent/particle concentration dependence to confirm that the data indeed reflects seeding by surface nucleation because this point affects the molecular interpretation of seeding kinetics. I also suggest figure 5 to be completely replaced by SI table 1 and SI fig 6 as they are the display items with the key data.

Version 1:

Reviewer comments:

Reviewer #1

(Remarks to the Author)

REVIEWER 1 ASSESSMENT (RESPONSE TO COMMENT #1):

The authors are correct that p3 has been detected in human brain and CSF by immunohistology or extracted as a monomeric peptide. However, it remains to be proven that p3 fibrils are present at significant levels in human brain tissue. While it is agreed that it is likely that these p3 fibrils are present in the human brain and it is logical to propose they would seed A β fibrils similarly to the submitted in vitro experiments, without establishing their actual presence in human brain, the clinical relevance of this work is still in questions. Methods for purifying aggregated/fibrillar A β from human tissue and seeding experiments have been described (PMC6820800, PMC9909698, PMC6326079), therefore, it is possible similar experiments with p3 could be done from human tissue or fluids.

REVIEWER 1 ASSESSMENT (RESPONSE TO COMMENT #2):

Acceptable response.

Reviewer #2

(Remarks to the Author)

All my concerns have been addressed in the revision.

Reviewer #3

(Remarks to the Author)

Upon careful reading of the revised manuscript and the authors' response to my previous comments, I think the authors have replied to all of my comments with the exception of the revisions in the new supplementary figure 1.

In the new supplementary table 1, the authors report on the mean values for the fibril widths and cross-over distances. However the helical pitch calculation is incorrect so the formula and the third column should be removed. The helical pitch is the length of the filaments with one complete helix turn, measured parallel to the helical axis. One needs to know the precise helical symmetry in order to estimate the pitch from the cross-over distance measurements so the helical pitch cannot be determined solely using the crossover distance. In my previous comment, what I suggested was that it would be very interesting to compare the measured crossover distance with the helical pitch (and therefore also the cross-over distance) values seen in the recently published cryo-EM models of patient brain derived Abeta40/42 filaments. In addition, I suggest a reference in the table legend of supplementary table 1 to the excellent new Supplementary Figure 7 which illustrates the full distributions seen in the data.

In addition to my comments, the authors have also added cell viability data using rat primary neuronal cells in this revision, which I think adds significantly to the biological significance of their biophysical data. Therefore, I fully support this manuscript to be published once Supplementary table 1 is corrected.

12th Nov 2024**Nature communication response NCOMMS-24-39090****Title: “The p3 peptides, A β _{17-40/42}, rapidly form amyloid fibrils that cross-seed with full-length A β , and the p3₄₂ annular oligomers form ion channel pores”**

Below is a point-by-point response to the reviewer’s comments.

Reviewer 1

In this manuscript the authors describe the characterization of the amyloidogenic potential of p3 fragments from APP. Historically, this has been considered to be the product of non-pathogenic (non-amyloidogenic) processing of APP. The authors report that the two major p3 peptides (40 and 42) can both efficiently form amyloid fibrils similar to full length A β peptides. These peptides also formed oligomers and annular structures like A β . Most importantly, the p3 peptides were able to seed the formation of A β peptide fibrils implicating them in Alzheimer’s disease pathogenesis. Further characterizations included the association of p3 aggregates with membranes, membrane disruption (calcium leakage), cellular toxicity, and calcium pore formation. This very interesting work shows the potential importance of these abundantly produced p3 peptides to Alzheimer’s disease pathogenesis as well as providing a plethora of important biochemical and physiological findings for them.

Major Concerns:

1) While these are exciting findings demonstrating many properties of p3 peptides and their effects on A β fibrillization, the clinical relevance is not yet fully determined. For this, the authors should provide direct evidence of p3 peptide fibrils isolated from human brain tissues. Most importantly, the demonstration of A β fibril seeding by p3 peptides derived from human brain tissue would go a long way to establishing the clinical relevance of this work.

RESPONSE: We thank the reviewer for the suggestions. In our introduction, paragraph 3, we cite several references that describe p3 peptide isolated from human brains, from diffuse and mature plaques in Alzheimer’s disease and Downs (Ref 11,12,14). We have now added further references indicating the p3 peptide is found in cerebrospinal fluid (Ref 13). We note that the α -secretase cleavage pathway is a well-established processing route for the amyloid precursor protein (Ref 8-10).

Seeding studies with brain derived human tissue using extracted p3 fibrils from a heterogeneous A β /p3 plaque might not be possible. This is because isolation of the

p3 peptide from A β plaques will result in the loss of the fibril structure. Certainly, this type of experiment would be a major challenge, but might be something for a future study. The presence of the p3 peptides in AD and Downs brain, together with our new data highlighting cytotoxicity in neuronal primary culture (Fig. 8), supports the p3 peptides clinical relevance. In addition, we have shown p3 will form oligomers and ion-channel pores across plasma membranes, and has significant cross-seeding properties with A β .

2) Fibril formation by p3 (40) has been previously reported. As well, seeding of A β fibrillization has previously been reported (PMID: 32412731). It was also reported that p3 can assemble with A β and alter its biological properties (PMID: 38236059). Therefore, some of the data presented in the manuscript are not entirely novel and the core concept of p3 peptide fibrillization and seeding has already been reported.

RESPONSE: We thank the reviewer for the comments. There have been some biophysical studies of p3 peptides, which we have been careful to cite, throughout the manuscript, please see for example paragraph 5, line 61. These studies are rather limited, there is no insight as to the mechanism that determines fibril assembly, which we have shown for both p₃₄₂ and p₃₄₀ in Figure 1 and supplementary Fig. 3,4. Significantly there is no convincing data of p3 seeding full-length A β . For these reasons the biophysical studies we show in our manuscript are a significant development worthy of nature communications, as is also indicated by the referees.

Minor Concerns:

i. Can thioflavin-T alter the fibrillization properties of peptides?

RESPONSE: We thank the reviewer for the comments. Thioflavin-T (ThT) is the gold-standard for monitoring amyloid fibril kinetics. Concerns that ThT might measurably impact fibril growth are unfounded. Our own carefully designed and executed studies show that amyloid fibril growth is unaffected by the presence of ThT, for a number of peptides. Please see (Ref 68). Furthermore, the same amount of ThT is used in all our kinetic studies, thus any effect of ThT will be the same for each condition, consequently differences in kinetic behaviour can be assumed to be independent of ThT. The TEM imaging, cytotoxicity, Ca(II) influx and electrophysiology studies are performed in the absence of ThT.

ii. Within the abstract and some places in the body of the text, data is used in the singular tense however data are plural.

RESPONSE: We thank the reviewer for the suggestions. We have gone through our manuscript carefully, checking for the datum/data (singular/plural) issue. The “data” (not datum) is correct in the abstract as it refers to multiple pieces of data.

Reviewer 2

The fundamental problem with the development of an effective therapy for Alzheimer's disease (AD) lies in the incomplete identification of the factor(s) responsible for its development. To date, our knowledge is based on longer, aggregation prone A β fragments (particularly A β 1-42 and 1-43) resulting from APP proteolytic processing and abnormally phosphorylated Tau protein. Additionally, it seems that only the oligomeric forms (not amyloid aggregates) of these peptides/proteins may play an important role in the development of the disease. This work focuses on the shorter A β fragments: A β 17-40 and 17-42, previously, perhaps prematurely, considered irrelevant in AD pathogenesis. The authors demonstrate that p3 peptides can form amyloid aggregates, cross-seed amyloidogenesis of compatible A β 1-40/42 counterparts and, more importantly, form cytotoxic oligomers (though slightly less toxic compared to A β 1-42/ 1-40). That is why the work seems to be important for exploring the mechanism of AD, albeit the comments mentioned below need to be addressed.

Key points:

1. In the Abstract (lines 21-23), the authors claim that: "The p3 peptide can cross-seed A β fibril formation, triggering oligomer and fibril formation in Alzheimer's disease, thus until now, their potential role in disease pathology has been largely misunderstood." while triggering oligomer formation by p3 peptide has not been directly demonstrated in the manuscript. I propose to rewrite this sentence in a more speculative form.

RESPONSE: We thank the reviewer for the suggestions. We have rewritten this point in the abstract to make it more speculative as suggested.

"In vitro, p3 peptides can cross-seed A β fibril formation, triggering oligomer production, thus a role for p3 peptides in disease pathology should not be overlooked."

2. The authors analysed the effects of p3 peptides solely on HEK293 cells. Immortalised human embryonic kidney cells, although widely used in in vitro studies since the 1970s, are not an optimal model of neural cells. It would be very interesting to see, at least, the cytotoxicity of p3 peptides in the primary culture of neurons (or equivalent experimental model).

RESPONSE: We thank the reviewer for the suggestions. We have used HEK293 cells because this is the cell-line of choice for patch-clamp studies. We wish to compare ion-channel pore conductivity; cell viability and calcium influx measurements using the same cell-line. We have now performed careful studies with neuronal primary cell culture as suggested. This significant additional study has been added to the manuscript, now shown in Fig 8. The p3 oligomers do indeed cause cytotoxicity in neuronal primary culture, similar to the A β oligomers and support our observations made with HEK269 cells.

3. From the description in the Results & Discussion and Materials & Methods sections, it is not clear in what configuration the patch clamp experiments were actually conducted. On page 25 (lines 484-486): “All patch clamp measurements were conducted using excised membrane in an inside-out configuration, thereby facilitating the exposure of the A β oligomers, to the extracellular surface of plasma membrane HEK293 cells, while under a voltage-clamp mode.” If the patch clamp experiments were in fact performed in an inside-out configuration, then it should be “intracellular surface of plasma membrane” instead of “extracellular surface of plasma membrane” .

RESPONSE: We thank the reviewer for the comments. The patch clamp experiments are set up so that A β and p3 oligomers are presented to the extracellular surface of the excised plasma cell membrane. The patch-clamp community calls this type of set-up an ‘inside-out’ configuration. This name is rather counterintuitive and so may confuse a reader not familiar with patch-clamp methods. We have rephrased this methods section to help with clarity.

4. For how many TEM images are the presented in Fig 3 the p3 peptide oligomers representative? Please, if possible, show the original TEM images of ring-shaped p3 oligomers.

RESPONSE: We thank the reviewer for the suggestions. The class averages in negatively stained annular oligomeric particles (Fig. 9) are typically 150 particles for each 2D-class average. We have added this detail to the figure caption. We have now also included some individual micrographs, (>30 were recorded), in the supplementary material with the annular structures highlighted as requested, supplementary Fig. S5.

Minor:

1. In Abstract (line 16), I suggest replacing “C-terminal lengths” with “C-terminal residues”.

RESPONSE: We thank the reviewer for the suggestions. In the abstract we have replaced C-terminal length with C-terminal residue as requested.

2. On page 7 (line 168) and page 19 (line 326), instead of "urinal acetate" it should be "uranyl acetate".

RESPONSE: We thank the reviewer for the suggestions. We have corrected the typo for uranyl acetate.

3. On page 10 (line 259) please replace “pre-fibular’ with “pre-fibrillar”.

RESPONSE: We have corrected the typo pre-fibrillar

4. TEM images in Fig. 3 should be presented at the same magnification, i.e., the oligomer images should be at the same magnification as the other images in the figure. And 5. The same as above applies to Supplementary Fig. 5, i.e., to enable comparison of the morphological features of the amyloid aggregates, the image of A β 40 fibrils should be presented at the same magnification as the other images (with scale bar of 100 nm).

RESPONSE: We thank the reviewer for the suggestions. We have adjusted the TEM images, Fig.3 and supplementary Fig.6 (was supplementary Fig.5), to match their magnification as suggested.

6. The citation style seems to be not uniform.

RESPONSE: We thank the reviewer for the suggestions. We have insured citation style is uniform.

Reviewer 3.

This manuscript by Tian et al. details a biophysical study on the amyloid aggregation of A β 17-40/42 peptides. There are currently controversial and confusing evidence on these so called p3 peptides in terms of their amyloid forming potentials and biological implications in human diseases. This study incorporates a wide set of biophysical approaches to confirm that both of the A β 17-40/42 peptides do form amyloid fibrils at least in vitro and offer excellent quality data on the assembly mechanism and membrane disruption potential of the species formed during assembly. Therefore, I think this study presents a significant step forward regarding our molecular and mechanistic understanding on the amyloid assembly of the p3 peptides. Therefore, I am very supportive of this manuscript and recommend that it should be published after revisions. Below are some comments which I hope the authors will find helpful in revising and improving this manuscript.

1. The key procedure that seems to have made a big difference compared to previous work and subsequently enabled all of the characterisation of the p3 amyloid assembly seems to be the additional SEC purification step. This is similar to historically how reproducibility for Abeta40/42 experiments was originally improved (e.g. Erik Hellstrand et al. 2010 <https://doi.org/10.1021/cn900015v>) Therefore, given the importance of this step, the result of this step should be better reported and example SEC trace should be presented in the supplementary information and compared to Abeta40/42 SEC experiments as this will help our community to reproduce and advance on experiments on p3 peptides .

RESPONSE: We thank the reviewer for the suggestions. We have carefully described SEC purification in the methods section. We have now added SEC traces for p3 peptides, supplementary Fig. 14. The p3 peptides absorbance at 280nm is weak, because there are no aromatic residues in the p3 peptide.

2. The evidence base that the formed structures are indeed cross-beta amyloid structures is already strong in this case. But for completeness I think the authors should still demonstrate the key tinctorial properties of amyloid in that the A β 17-40/42 structures they form in vitro also bind Congo red and produce birefringence

RESPONSE: We thank the reviewer for the suggestions. The evidence that p3 forms amyloid fibrils is overwhelming, with very clear amyloid fibrils imaged, as well as clear prefibrillar curvilinear oligomers and annular oligomers. Circular dichroism indicates fibrils with beta-sheet content. Furthermore, fibril-growth curve kinetics fits to a classic secondary nucleation process. In addition, p3 peptides have characteristic self-seeding properties. Congo-red is a less specific dye that is occasionally used to image plaques *ex vivo*, however we do not feel it is warranted here.

3. The ThT kinetics data show in the manuscript are of excellent quality. However, given the known low reproducibility of these experiments, the authors should give more details of the variations observed. In the figure legends they report 3 replicates were recorded, this should be clarified as technical replicates. The batch-to-batch reproducibility of the series (e.g. on t50, gamma, etc) should be examined and commented. The exact type of plates used should be reported with product number details and the reading programme with the reading interval details should be given as all of these factors affect the reproducibility considerably.

RESPONSE: We thank the reviewer for the suggestions. We have added further details of the ThT kinetics data, which are technical replicates, see Fig. 2 and Fig. 4 captions. We have also added additional details regarding the kinetic measurements including the well plate type, reading interval, and other details in the methods section (line 382-385).

4. The authors reported the “node-to-node period” and the width of their fibril structures. How this was carried out is not currently described in the manuscript but should be reported. The “node-to-node period” the authors mention should be changed to cross-over distances, and I also suggest the authors to indicate both the cross-over distance and the helical pitch length reported in the cryo-EM structures of Abeta amyloid (e.g. in a SI table or visually in fig 3C) to help the readers to compare.

RESPONSE: We thank the reviewer for the suggestions. The cross-over distances were measured manually by image-J, well resolved fibrils, typically >300 fibrils were

measured for each peptide. We have now used the phrase “crossover-point” rather than “node-to-node” to describe the periodicity in the fibril twists. We have also calculated the helical pitch-length for the four peptides, shown as supplemental Table 1.

5. The EM images and the distribution of cross-over distances and widths reported in fig 3 clearly indicate that the samples are highly polymorphic. The distributions are, in fact, showing that the differences between all four samples are substantial and not “similar” as the authors state on page 5. The authors quote in the text mean value \pm standard deviation and this is very misleading as the casual readers will incorrectly infer similarity since the mean \pm standard deviation bounds overlap. However, in population analysis of polymorphic amyloid populations, these distributions are very different (e.g. one can have completely different distributions but very similar mean values, see Aubrey 2020 et al <https://doi.org/10.1038/s42004-020-00372-3>). Therefore, the authors should revise this section. The ranges of values should be quoted as ranges (e.g. the cross-over distances observed ranges from x to y). While the violin plots are visually nice, the distributions should also be plotted as normal histograms in the supplementary so that the readers can evaluate the differences and see the n-value used in the analysis. There should be also a short discussion or note on the clearly polymorphic nature of these samples in this section.

RESPONSE: We thank the reviewer for the suggestions. Although the fibrils are polymorphic, the ranges of twists, group very clearly by truncation in the C-terminus and not the N-terminus. Please see supplementary Fig 7, supplementary Table 1, and also Figure 3. We have rephrased this point in the manuscript. We have shown the cross-over points as histograms for the 4 peptides, in a new supplementary Fig. 7, we also describe the ranges of twist-periodicity as well as mean values. The precise structure of these fibrils is yet to be determined.

6. The cross-seeding data are very interesting. I agree with the authors that structural difference in the seeds most likely is the reason for the pattern observed in fig 5 (and SI table 1 and SI fig 6). However, the caveat is that under the conditions employed by the authors, the dominating seeding mechanism seems to be surface nucleation rather than templated elongation because even at high concentration (10%) seeds the lag phase is still present (the ability by low amounts of seed to remove lag phase is one of the criteria used to confirm seeding by templated elongation, see Koloteva-Levine et al. 2021, <https://doi.org/10.1073/pnas.2104148118>). To clarify on this important mechanistic point, I suggest the authors either find and carry out the cross-seeding experiments under condition where self-seeding is dominated by templated growth, or to carry out seed monomer equivalent/particle concentration dependence to confirm that the data indeed reflects seeding by surface nucleation because this point affects the molecular interpretation of seeding kinetics. I also suggest figure 5 to be

completely replaced by SI table 1 and SI fig 6 as they are the display items with the key data.

RESPONSE: We thank the reviewer for the suggestions. It is well understood that self-seeding, (with 10% fibril seeds) will circumvent primary nucleation. However, with a 10% seed, Fig. 4 show that lag-times are reduced (with a small impact on the slope) but even for self-seeding, 10% fibril seeds do not reduce lag-times to zero, under the conditions used here.

Whether the cross-seeding phenomena observed is driven by fibril lateral surface catalysed secondary nucleation or templating at the ends of seeding fibrils is interesting. We have added the following discussion to the manuscript.

Line 162-166: “It is notable the reduction in lag-times observed for cross-seeding is comparable in magnitude to the A β and p3 self-seeding kinetics, perhaps suggesting that the molecular processes are similar. It remains to be established if the cross-seeding phenomena is driven by fibril surface-catalysed secondary-nucleation or templating from the ends of seeding fibrils. In the case of self-seeding both lateral surface-catalysed secondary-nucleation and templating elongation occurs. Koloteva-Levine et al reports an excellent analysis of these two molecular behaviours⁵⁹.”

We have now read the excellent manuscript by Koloteva-Levine et al mentioned by the referee, and incorporated this in to our discussion. This elegant approach would represent a completely new line of enquiry, beyond the scope of this manuscript, which already contains ten figures and fourteen supplementary figures.

We have reworked Fig. 5 as suggested with related data presented in supplementary Fig. 8. The tabulated data has been presented as a percentage difference in seeded t(50) relative to self-seeding. The ‘raw data’ remains as a supplementary Fig. 8, as a summary of this data is in Fig 4. In addition, the t50 values for unseeded, self-seeding and cross-seeding conditions are added as a supplementary Table 2.

We believe we have addressed the referee’s comments in full and hope the manuscript can now be excepted for publication.

Yours sincerely,

John Viles

14th Dec 2024**Nature communication response NCOMMS-24-39090A****Title: “The p3 peptides, A β _{17-40/42}, rapidly form amyloid fibrils that cross-seed with full-length A β , and the p3₄₂ annular oligomers form ion channel pores”**

Below is a point-by-point response to the additional reviewer’s comments.

Reviewer 1

We thank the referee for their comments and pointing us to some publications involving isolation of brain derived amyloid fibrils. We are pleased the referee agrees it is not unreasonable to suggest even low levels of p3 fibrils may have a profound impact on cross-seeding A β ₄₀ and A β ₄₂. We agree, the next step should be to extract p3 fibrils from brain derived plaques. We have reviewed the manuscript carefully and toned down any *in-vivo* conclusions drawn from our *in-vitro* studies.

Reviewer 3.

We thank the referee for their comments. We have now revised the supplemental table S1 as requested and removed references to helical-pitch-length. We have also flagged supplemental figure 7 in the table 1 caption. We want to avoid direct comparison with helical crossover distances of published *in vivo* cryo-EM studies. This is because there is considerable variation and polymorphism within the same brain derived or *in vitro* sample. Fibril polymorphism is very interesting and we have added two references on this topic (line 124).

We have attached a version of the manuscript where the second set of revisions are highlighted.

Yours sincerely,

John Viles